# Thermal Modelling Utilizing Multiple Experimentally Measurable Parameters

Anosh Mevawalla [1],*, Yasmin Shabeer [1], Manh Kien Tran [1], Satyam Panchal [2], Michael Fowler [1] and Roydon Fraser [2]

1 Chemical Engineering Department, University of Waterloo, 200 University Avenue West, Waterloo, ON N2L 3G1, Canada

2 Mechanical and Mechatronic Engineering Department, University of Waterloo, 200 University Avenue West, Waterloo, ON N2L 3G1, Canada

* Correspondence: amevawal@uwaterloo.ca; Tel.:+1-519-274-5815

**Abstract:** This paper presents three equivalent thermal circuit models with multiple input parameters, namely, the state of health (SOH), state of charge (SOC), current and temperature. Typical physiochemical models include parameters such as porosity and tortuosity, which are not easily experimentally available; this model allows for model parameters such as the internal impedance to be easily estimated using more practical inputs. The paper models the internal impedance resistance of a $LiFePO_4$ battery at five different ambient temperatures (5, 15, 25, 35, 45 °C), at three different discharge rates (1C, 2C, 3C) and at three different SOHs (90%, 83%, 65%). The internal impedance surface fit experimental measurements with a Pearson coefficient of 0.945. Three thermal models were then created that implemented the internal resistance model. The first two thermal models were 0D models that did not include the influence of the thermal conductivity of the battery. The first model assumed simple heating through internal resistance and convection energy loss, while the second also included the Bernardi Reversible heat term. The final third model was a 2D model that included all previous heat source terms as well as tab heating. The 2D model was solved using a simple Euler method and finite center difference. The $R^2$ values for the 0D thermal models were 0.9964 and 0.9962 for the simple internal resistance and reversible heating models, respectively. The $R^2$ value for the 2D thermal model was 0.996.

**Keywords:** lithium ion battery; heat transfer; surface methods; equivalent circuit model; physiochemical model; thermal model





## 1. Introduction

There are many different types of models when modeling lithium ion batteries, the most common being equivalent circuit models, electrochemical models and artificial intelligence data-driven approaches such as recurrent neural networks [1–6].

Electrochemical physical models primarily consider ohms law in electrode, mass transfer in electrode and electrolyte and intercalation/deintercalation kinetics [7]. Electrochemical models are robust but slow even when considering the simplifications of extended single particle model (ESPM). ESPM models simplify anode and cathode modeling by using only a single particle at each electrode [2–4]. Electrochemical models simulate the effects of diffusion and charge transfer on the cell's voltage [8]. They are also used to predict SOH by modeling SEI growth, lithium plating and cracking with active material loss. The entropic heat parameter is of particular interest and is typically modeled simply as a function of the SOC but is also a function of temperature to some extent [7].

Equivalent circuit models are known for their easy implementation, simple structure and easy integration with Kalman filter estimation [8,9]. They do, however, require large data tables modeling parameters as functions of typically SOC and temperature. They are found particularly to be weak with regard to SOH changes [10], and this paper seeks to

alleviate the issue through resistance estimation which also minimizes the data table size. Overall, equivalent circuit models have low computational effort when fitting parameters while having physical meaning.

Long-term short-term recurrent neural network models have no physical meaning but require less effort to create and fit while electrochemical models such as ESPM require less data than neural networks but more calibration effort as more three sets of tests are needed [10]. Electrochemical models also require less parameters overall.

Electric vehicle modules need to maintain the battery module temperature between 25 °C and 40 °C with the temperature distribution less than 5 °C to prevent capacity fading and the threat of thermal runaway [11]. Lithium ion cells with high nickel content are prone to thermal reactivity and exothermic decomposition as reported by Liang et al. [10]. The maximum operating temperature for most lithium ion cells is 50 °C while, at temperatures above 60 °C, there is threat of thermal runaway [12,13]. It is known that, at temperatures above 80 °C, the SEI layer breaks down [14]. Dissolution of the metal ions at the cathode is seen from the 25 °C to 90 °C region for NMC cells [15]. We see Li-ion batteries can fail under conditions of abuse, such as overcharge, overdischarge, physical penetration, short-circuit, overheating, accelerated penetration, etc. [16,17] and accurate battery modeling is key to preventing this.

This paper focuses on an equivalent circuit model approach that incorporates physio-chemical theory into developing a nonlinear equation for the internal resistance. Once the nonlinear model for the internal resistance is built we use a simple thermal model to simulate heating effects both from the internal resistance and secondly from the reversible heat. The thermal 2D model also incorporates tab resistances as an additional heat source term. Physiochemical models do not allow a practical method to estimate battery performance as they require externally non-measurable parameters. This paper seeks to alleviate this issue by modeling the total internal resistance, overpotential and heat produced as a function of easily measured practical parameters that are related to fundamental physical parameters. The paper models the internal impedance resistance of a $LiFePO_4$ battery at 5 different ambient temperatures (5, 15, 25, 35, 45 °C), at 3 different discharge rates (1C, 2C, 3C) and at 3 different SOHs (90%, 83%, 65%). The experimental setup consisted of a 20-Ah prismatic $LiFePO_4$ A123 lithium-ion battery with the cell specifications are shown in Table 1.

**Table 1.** Prismatic 20-Ah Lithium-ion ($LiFePO_4$) cell specifications. Adapted from Ref. [7].

| Specification | Value | Unit |
|---|---|---|
| Material for electrolyte | Carbonate based | - |
| Material for anode | Graphite | - |
| Material for cathode | $LiFePO_4$ | - |
| Voltage (nominal) | 3.3 | V |
| Dimensions | 7.25 (t) × 160 (w) × 227 (h) | mm |
| Capacity of the cell (nominal) | 72,000 | C |
| Discharge power | 1200 | W |
| Energy (nominal) | 234,000 | J |
| Specific energy | 471,600 | J/kg |
| Energy density | 889.2 | $J/m^3$ |
| Operating temperature | −30 to 55 | °C |
| Storage Temperature | −40 to 60 | °C |
| Mass of the cell | 0.541 | Kg |
| Specific power | 2400 | W/kg |
| Internal resistance | $5 \times 10^{-4}$ | Ω |
| Volume | $2.63 \times 10^{-4}$ | $m^3$ |
| Storage temperature | −40 to 60 | °C |
| Number of cycles | Min. 300, approx. 2000 | Cycles |
| Max Discharge Current | 300 | A |
| Max Charge Current | 300 | A |

The cooling method was natural convection through air cooling, with the battery being placed vertically in a stand inside the thermal chamber. The test bench consisted of four components: (1) Maccor battery tester; (2) thermal chamber; (3) 8-channel USB thermal couple; (4) computer.

The temperature was measured using T-type thermocouples. The thermocouples were connected to the NUC computer through a USB data logger. The locations of the thermocouples are shown in Figure 1. The back of the battery has the thermocouples in the same positions as the front of the battery with thermocouple 5 being on the anode, 6 being on the cathode, 7 in the center and 8 on the bottom.

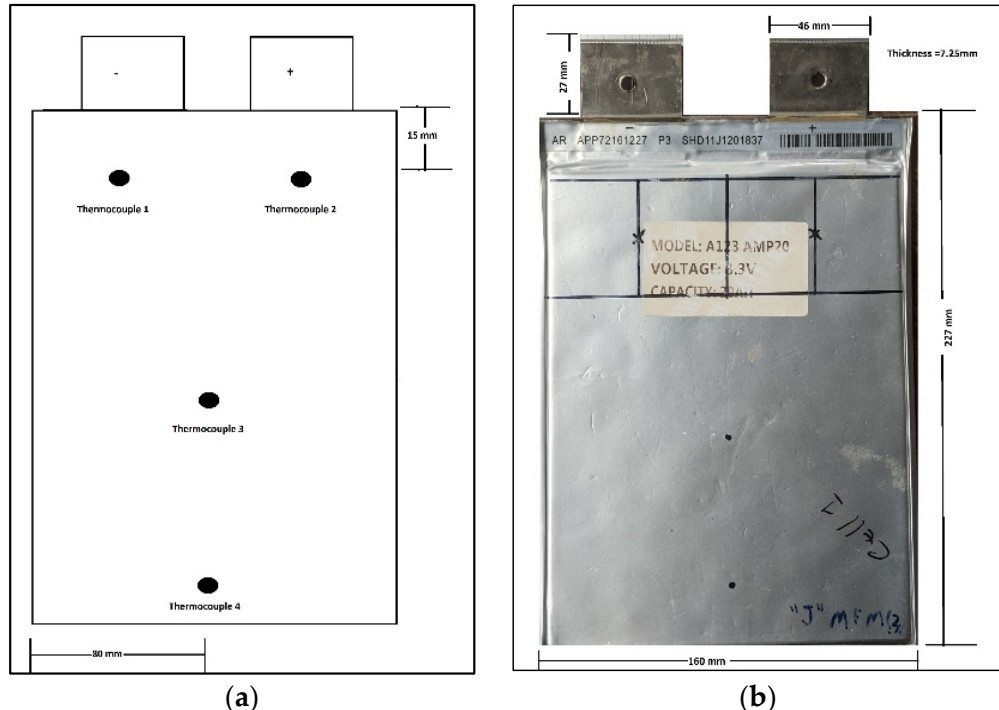

(**a**)            (**b**)

**Figure 1.** Thermocouple locations and physical dimensions. (**a**) Thermocouple locations; (**b**) Physical dimensions. Adapted from ref. [7].

The average surface temperature was calculated by using a Voronoi diagram, which allows us to obtain the points closest to each individual thermocouple and assign it to the area for that thermocouple. The Voronoi diagram is show in Figure 2.

The points shown in Figure 2 show the locations of the thermocouples on the battery surface. The use of the Voronoi diagram ensures that the area assigned to each thermocouple is the area that is truly closest to the sensor's point on the surface and is not simply an arbitrary area assigned to that weighted measurement. By doing this, the appropriate magnitude of surface area is assigned to each thermocouple's experimental measurement. This confirms the single point average surface area temperature is a legitimate measurement of the true battery surface temperature.

The battery's average surface temperature is then calculated by multiplying the thermocouple measurement with the area for the location, summing up the weighted measurements and dividing by the total area. It is the area weighted average for the battery. The equation is shown below:

$$T_{Average} = \sum_{i=1}^{i=8} \frac{T_{i\ measured} \times Area_i}{Total\ Area} \tag{1}$$

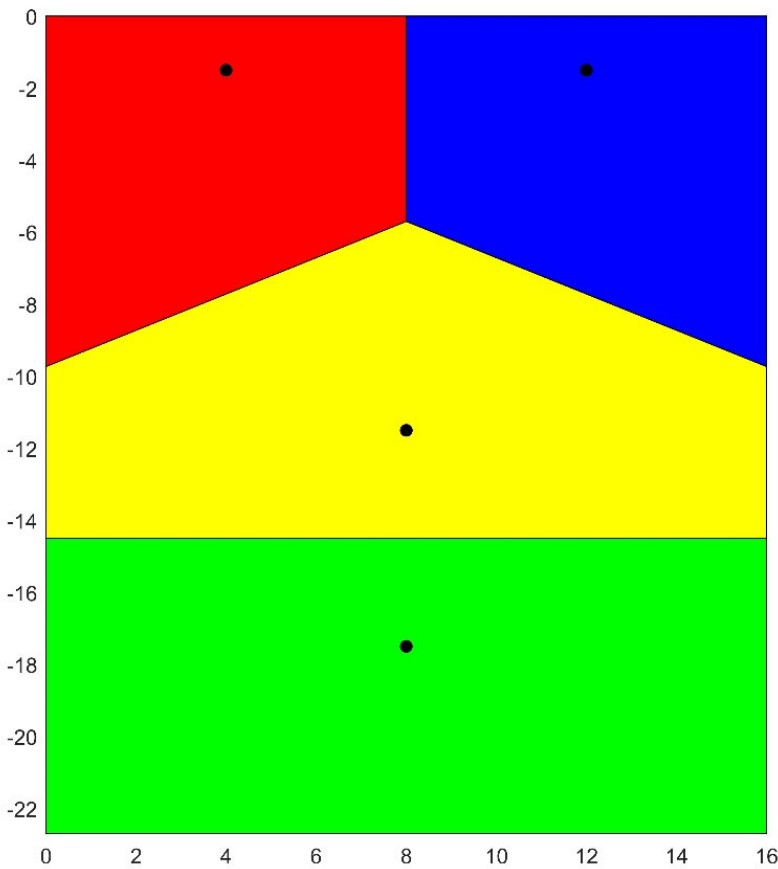

**Figure 2.** Voronoi Diagram with Thermocouple Locations shown as black dots.

The batteries were charged at constant current-constant voltage (CCCV) at 1C and allowed to rest for 2 h to equilibrate to ambient temperature set in the thermal chamber. The thermal couples then recorded the behavior of the battery at three different constant current discharge rates: 1C, 2C and 3C. The five different ambient temperatures that were tested were at 5 °C, 15 °C, 25 °C, 35 °C, and 45 °C. Three batteries at 90.575%, 83.435%, and 65.025% SOH were tested at the conditions described. The internal resistance was then modeled under these operating conditions.

## 2. Internal Resistance

Internal resistance is one of the most important battery parameters when modeling lithium ion batteries. The typical time constants for RC pairs can range from a few seconds to hundreds of seconds [10]. In addition to the internal resistance being the primary cause of heat generation, since the time constants for a 1RC model are small and our experiments are carried out at constant current, we instead used a single Rint model for the simulation.

Characteristic energy losses in the battery include the mass concentration loss, the activation loss and the ohmic loss. All these energy losses lead to heat production in the battery in addition to the reversible heat of reaction. The reversible heat of reaction is not included in the internal resistance term as it does not contribute to an overpotential. The $R_{int}$ model can be parameterized directly from the experimental data and is very efficient as there is no need to evaluate the differential equations. However, the model is unable to provide a simulation of transient or time-variant behavior.

$$V_L = V_{ocv} - IR_{int} \tag{2}$$

The following model is then derived to model the internal resistance as a function of the SOH, SOC, T and C rate.

$$R_{int} = A_1 + (A_2 + 1)SOH * A_3 + (A_2 + 1)SOH * T * SOH * A_4 * \frac{asinh\left(\frac{I*SOC}{A_5}\right)}{I*SOC}$$
$$+ \frac{\left(A_6*ln((A_2+1)SOH*I*SOC)+A_7+\frac{A_8}{T}\right)}{I*SOC} + A_9 * (A_2 + 1)SOH * ln\left(\frac{1}{SOC}\right) * exp\left(\frac{A_{10}}{T}\right) \quad (3)$$

The proposed model is a 10 parameter nonlinear equation that has been fit using MATLAB's nlinfit function. The Pearson coefficient for the 45 experiments (3 C-rates × 3 SOHs × 5 temperatures) was 0.945. The parameter values are presented in Table 2.

**Table 2.** Parameter values of Rint model for 20 Ah Lithium-ion (LiFePO$_4$) cell.

| Parameter | Value |
|---|---|
| $A_1$ | $465 \times 10^{-3}$ |
| $A_2$ (soh $\leq$ 0.75) | 2196 |
| $A_2$ (soh > 0.75) | 0 |
| $A_3$ | $-1.463 \times 10^{-5}$ |
| $A_4$ | 3.596 |
| $A_5$ | $6.51 \times 10^7$ |
| $A_6$ | $4.232 \times 10^{-3}$ |
| $A_7$ | $-0.309$ |
| $A_8$ | 97.6 |
| $A_9$ (soc $\leq$ 0.35) | 52,690 |
| $A_9$ (soc > 0.35) | 0 |
| $A_{10}$ | $-7643$ |

The model is shown as multiple surface plots in Figure 3. The figures are of internal resistance surfaces at the three different experimental C rates, with each figure itself being for the specified SOH. We note the $A_2$ terms are a function of the SOH, while the $A_9$ term is a function of the SOC. The last term is noted to be a compound effect of the SOC and the SOH with the temperatures effect being inside an exponential. The internal resistance is seen to increase dramatically at low SOC. For SOHs above 75% and SOCs above 35%, we see there is no linear effect with temperature and simply an inverse relation with the internal resistance. We note that the diffusion coefficient for the graphite anode is a function of the exponential of the reciprocal of temperature from the temperature range of 120 °C to −50 °C extrapolating outside of which we assume a constant effect [7]. We note that, according to the Modified Butler–Volmer equation, we would expect to see a term dependent on the asinh of the current and the SOC along with the temperature divided by the current and SOC as seen in the third term.

We note that the internal resistance is seen to decrease with an increase in the C rate in this current range, which is still not near the current limit for the cell. The overall energy loss and the overpotential however increases as C-rate increases. The general increase in impedance at low temperatures is mainly attributed to the lowered diffusion coefficient of the electrolyte. The experimental data are shown up to 5 °C. At low SOC, mass transfer losses play an increased role and the impedance shows an exponential increase.

The residuals for the internal impedance are shown in Figure 4 below. The residuals show some trends with both the SOC and the temperature and are not randomly distributed along the vertical axis. They are however distributed fairly evenly along the 0 error line shown in red.

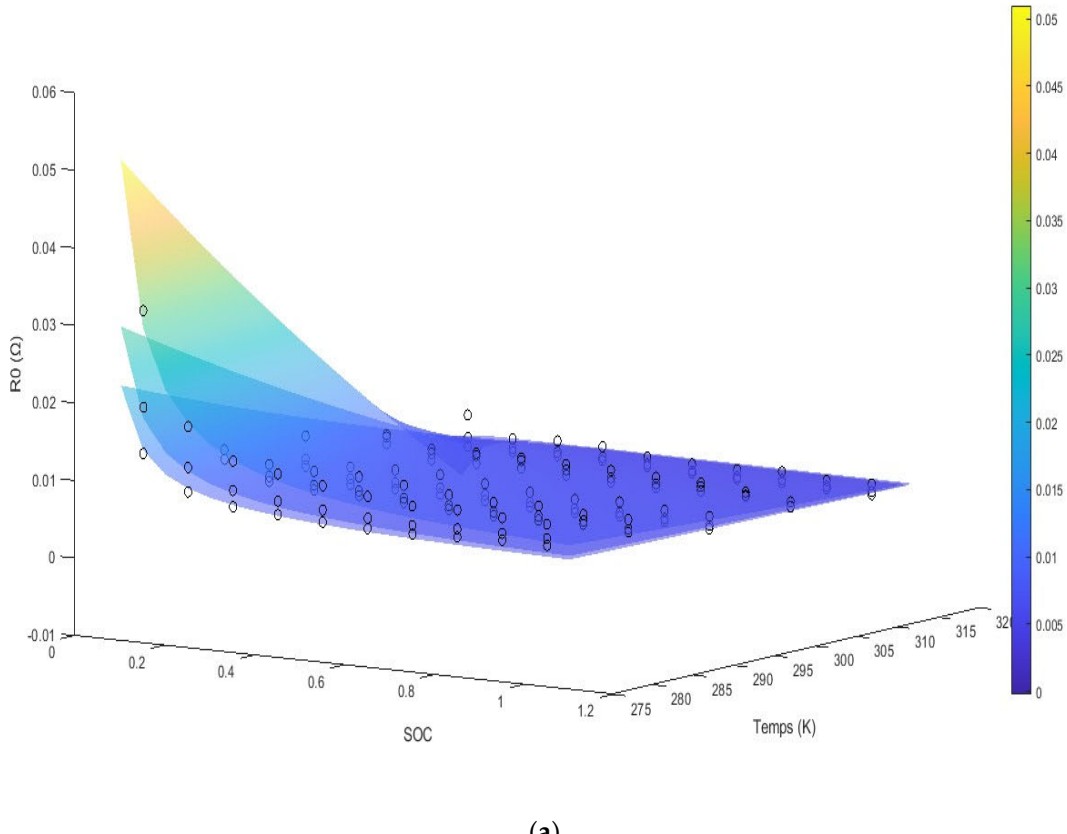

(**a**)

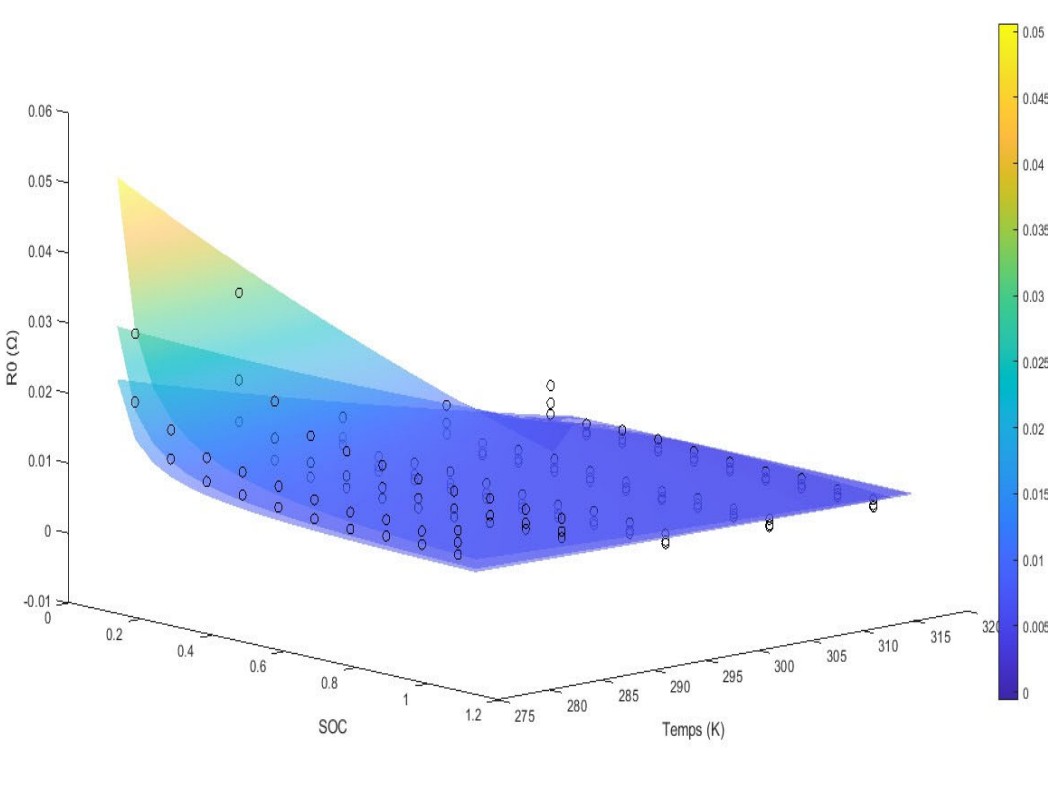

(**b**)

**Figure 3.** *Cont*.

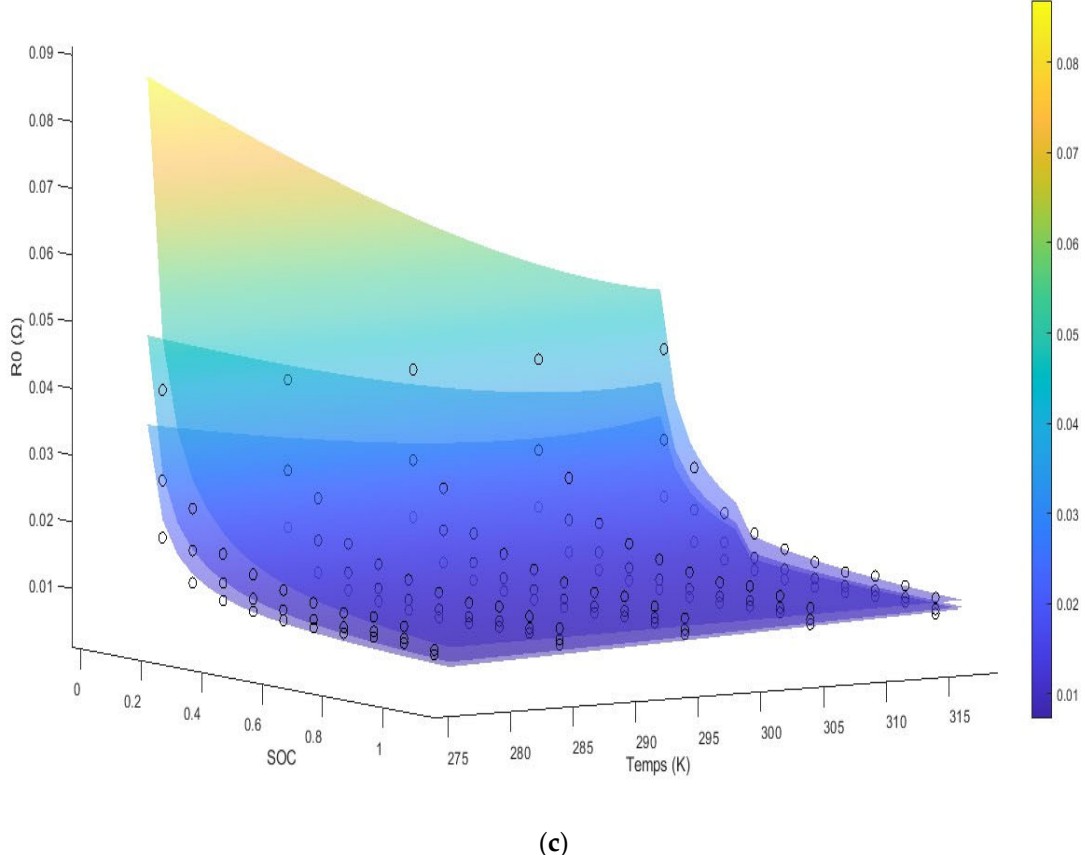

(**c**)

**Figure 3.** (**a**–**c**): Surface plots of Internal Impedance modeled from Equation (3) with each surface being at the shown C-rate and each figure for the stated SOH. (**a**) 90.575% SOH; (**b**) 83.435% SOH; (**c**) 65.025% SOH. (Sample Experimental Internal Resistances shown as circles).

The plots show some trend indicating that the internal resistance is a function of SOC and Temperature (or a variable that covaries with them) in a nonlinear way that is not included in model. We pay attention to the trend seen at 83% SOH and 3C to note that the SOC term in the model is best expressed conditionally with an interaction with both the SOH and current. We note that the current plays a significant role comparing the residuals. Once again at 83% SOH and 3C we note a nonlinear trend in temperature not captured by the model. The residuals also so heteroscedasticity which typically indicates the lack of an input variable interaction term with the independent variable. The interaction term at lowest SOH 65% and 1C shows heteroscedasticity that converges with SOC while at higher C rates the residual variance decreases then increases again. At 3C and 65% SOH we see the residuals vary linearly with SOC indicating the lack interaction term that is linear at high C and conditional on both SOH and C rate.

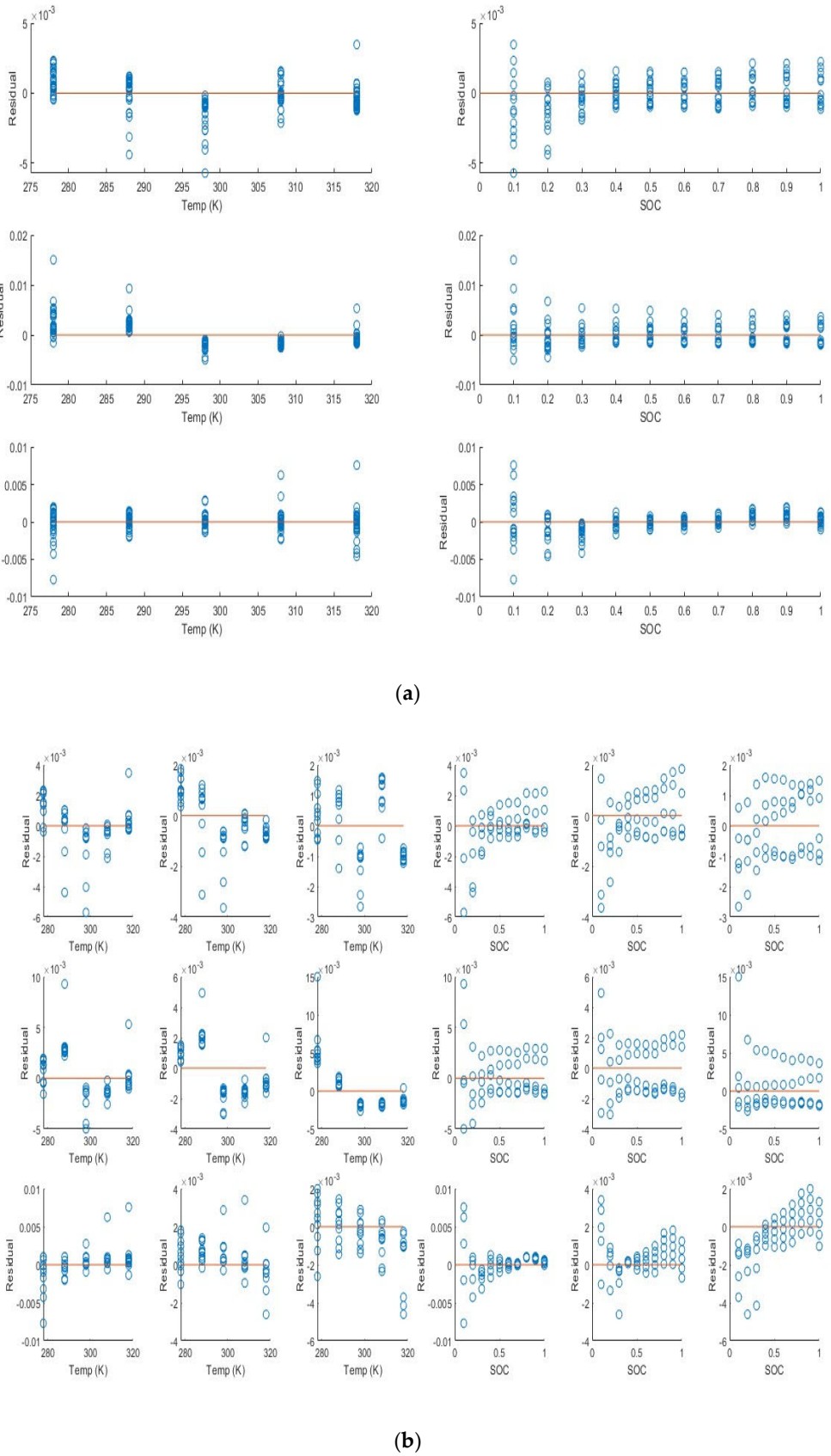

(**a**)

(**b**)

**Figure 4.** (**a**) Residual plot of Internal Impedance modeled from Equation (3) with residuals plotted against Temperature and SOC (**b**) Residual plot of Internal Impedance modeled from Equation (3) with residuals plotted against Temperature and SOC with each C-rate on a separate plot.

### 3. 0D Thermal Models

The first thermal model is a simple thermal model that uses the parameters in Table 2 and Equation (3) to solve the differential equation below:

$$I^2 R_{int} - hA(T - T_{amb}) = \rho c_p \frac{dT}{dt} \qquad (4)$$

The second thermal model includes an additional heat source term, the heat of reaction, and is the 0D form of the Bernardi heat equation:

$$I^2 R_{int} + IT\frac{dV_{ocv}}{dT} - hA(T - T_{amb}) = \rho c_p \frac{dT}{dt} \qquad (5)$$

The values of the open circuit voltage derivative with respect to temperature are given as a function of SOC in Tables 3 and 4 for the cathode and anode, respectively.

**Table 3.** Open circuit partial derivative with respect to temperature for the iron phosphate cathode.

| SOC | dEq/dT | SOC | dEq/dT | SOC | dEq/dT |
|---|---|---|---|---|---|
| 1 | $8.62 \times 10^{-6}$ | 0.69 | $-1.97 \times 10^{-5}$ | 0.38 | $-7.13 \times 10^{-5}$ |
| 0.99 | $8.62 \times 10^{-6}$ | 0.68 | $-2.15 \times 10^{-5}$ | 0.37 | $-7.29 \times 10^{-5}$ |
| 0.98 | $2.88 \times 10^{-5}$ | 0.67 | $-2.34 \times 10^{-5}$ | 0.36 | $-7.47 \times 10^{-5}$ |
| 0.97 | $4.27 \times 10^{-5}$ | 0.66 | $-2.54 \times 10^{-5}$ | 0.35 | $-7.66 \times 10^{-5}$ |
| 0.96 | $5.15 \times 10^{-5}$ | 0.65 | $-2.74 \times 10^{-5}$ | 0.34 | $-7.87 \times 10^{-5}$ |
| 0.95 | $5.64 \times 10^{-5}$ | 0.64 | $-2.94 \times 10^{-5}$ | 0.33 | $-8.10 \times 10^{-5}$ |
| 0.94 | $5.81 \times 10^{-5}$ | 0.63 | $-3.15 \times 10^{-5}$ | 0.32 | $-8.33 \times 10^{-5}$ |
| 0.93 | $5.75 \times 10^{-5}$ | 0.62 | $-3.37 \times 10^{-5}$ | 0.31 | $-1.40 \times 10^{-4}$ |
| 0.92 | $5.51 \times 10^{-5}$ | 0.61 | $-3.58 \times 10^{-5}$ | 0.3 | $-1.40 \times 10^{-4}$ |
| 0.91 | $5.15 \times 10^{-5}$ | 0.6 | $-3.79 \times 10^{-5}$ | 0.29 | $-1.50 \times 10^{-4}$ |
| 0.9 | $4.70 \times 10^{-5}$ | 0.59 | $-4.00 \times 10^{-5}$ | 0.28 | $-1.50 \times 10^{-4}$ |
| 0.89 | $4.21 \times 10^{-5}$ | 0.58 | $-4.21 \times 10^{-5}$ | 0.27 | $-1.50 \times 10^{-4}$ |
| 0.88 | $3.69 \times 10^{-5}$ | 0.57 | $-4.41 \times 10^{-5}$ | 0.26 | $-1.60 \times 10^{-4}$ |
| 0.87 | $3.18 \times 10^{-5}$ | 0.56 | $-4.61 \times 10^{-5}$ | 0.25 | $-1.60 \times 10^{-4}$ |
| 0.86 | $2.67 \times 10^{-5}$ | 0.55 | $-4.80 \times 10^{-5}$ | 0.24 | $-1.70 \times 10^{-4}$ |
| 0.85 | $2.19 \times 10^{-5}$ | 0.54 | $-4.98 \times 10^{-5}$ | 0.23 | $-1.70 \times 10^{-4}$ |
| 0.84 | $1.74 \times 10^{-5}$ | 0.53 | $-5.15 \times 10^{-5}$ | 0.22 | $-1.70 \times 10^{-4}$ |
| 0.83 | $1.32 \times 10^{-5}$ | 0.52 | $-5.32 \times 10^{-5}$ | 0.21 | $-2.10 \times 10^{-4}$ |
| 0.82 | $9.43 \times 10^{-6}$ | 0.51 | $-5.47 \times 10^{-5}$ | 0.2 | $-1.70 \times 10^{-4}$ |
| 0.81 | $5.98 \times 10^{-6}$ | 0.5 | $-5.62 \times 10^{-5}$ | 0.19 | $-1.70 \times 10^{-4}$ |
| 0.8 | $2.86 \times 10^{-6}$ | 0.49 | $-5.75 \times 10^{-5}$ | 0.18 | $-1.70 \times 10^{-4}$ |
| 0.79 | $5.61 \times 10^{-8}$ | 0.48 | $-5.88 \times 10^{-5}$ | 0.17 | $-1.70 \times 10^{-4}$ |
| 0.78 | $-2.48 \times 10^{-6}$ | 0.47 | $-6.00 \times 10^{-5}$ | 0.16 | $-1.70 \times 10^{-4}$ |
| 0.77 | $-4.79 \times 10^{-6}$ | 0.46 | $-6.12 \times 10^{-5}$ | 0.15 | $-1.70 \times 10^{-4}$ |
| 0.76 | $-6.91 \times 10^{-6}$ | 0.45 | $-6.24 \times 10^{-5}$ | 0.14 | $-1.70 \times 10^{-4}$ |
| 0.75 | $-8.88 \times 10^{-6}$ | 0.44 | $-6.35 \times 10^{-5}$ | 0.13 | $-1.70 \times 10^{-4}$ |
| 0.74 | $-1.08 \times 10^{-5}$ | 0.43 | $-6.47 \times 10^{-5}$ | 0.12 | $-1.60 \times 10^{-4}$ |
| 0.73 | $-1.26 \times 10^{-5}$ | 0.42 | $-6.58 \times 10^{-5}$ | 0.11 | $-1.60 \times 10^{-4}$ |
| 0.72 | $-1.43 \times 10^{-5}$ | 0.41 | $-6.71 \times 10^{-5}$ | 0.1 | $-1.60 \times 10^{-4}$ |
| 0.71 | $-1.61 \times 10^{-5}$ | 0.4 | $-6.84 \times 10^{-5}$ | 0 | $-3.10 \times 10^{-4}$ |
| 0.7 | $-1.78 \times 10^{-5}$ | 0.39 | $-6.98 \times 10^{-5}$ | | |

**Table 4.** Open circuit partial derivative with respect to temperature for the graphite anode.

| SOC | dEq/dT |
|---|---|
| 0 | $3.00 \times 10^{-4}$ |
| 0.17 | 0 |
| 0.24 | $-6.00 \times 10^{-5}$ |
| 0.28 | $-1.60 \times 10^{-4}$ |
| 0.5 | $-1.60 \times 10^{-4}$ |
| 0.54 | $-9.00 \times 10^{-5}$ |
| 0.71 | $-9.00 \times 10^{-5}$ |
| 0.85 | $-1.00 \times 10^{-4}$ |

We can see the open circuit voltage derivative with respect to temperature is expressed as a function of SOC and we note that the reversible heat of reaction is an exact differential and is not path-dependent. Additionally, the change in sign of the differential indicates a change in entropy and that the proposed reaction's number of moles of products versus reactants is not constant.

The model parameters are presented in Table 5. Along with the cell specifications in Table 1, these parameters are enough to solve for the simulation equations.

**Table 5.** 0D Model Parameters.

| Parameter | Value |
|---|---|
| h | $30 \text{ W/m}^2$-K |
| $C_p$ | 1399.1 J/Kg-K |

The simulated temperatures are shown in Figure 5. The models show good agreement with experimental results. The Pearson coefficients for the reversible model are seen to be lower than for the simpler model with no reversible heat term when the temperatures are 5 °C and 15 °C for the experimental SOHs of 83% and 90%. For the experimental temperatures of 25 °C and over, we see the reversible heat model is more accurate at the experimental SOHs of 83% and 90%. For the SOH of 65%, we see the non-reversible model to be more accurate only at the lowered 5 °C margin.

Figure 5 shows that the Rint model can accurately be used to model the temperature at all 45 experimental conditions. The model is seen to be robust yet still quite accurate at modeling the average battery temperature from all 8 thermocouples. The effect of the reversible heat is seen to be cooling at high SOCs with a steep heating effect at lower SOCs. This indicates the entropy is negative at low SOCs and is positive and high SOCs. This is a thermodynamic oddity. The Pearson coefficients for all 0D models are once again presented below in Table 6.

The highest accuracy Pearson values are seen to occur at 288 K, 65% SOH and 54 Amp current for the non-reversible model and at 278 K, 91% SOH and 54 Amp current for the Reversible model. Exothermic reactions are supposed to have equilibrium constants that decrease with increases in temperature indicating that at low temperatures the reversible heat should be larger. With regard to SOH, both reversible and non-reversible models have the highest correlation coefficients at 83% followed by 91% and finally 65% SOH displaying the least accuracy in large part due to rapid degradation and changes in the SOH itself. Overall, we see, for the non-reversible heat model, the accuracy increases from 278 K to 288 K and then decreases with temperature, while, for the reversible heat model, the Pearson coefficient increases from 278 K to 308 K and then decreases from 308 K to 318 K.

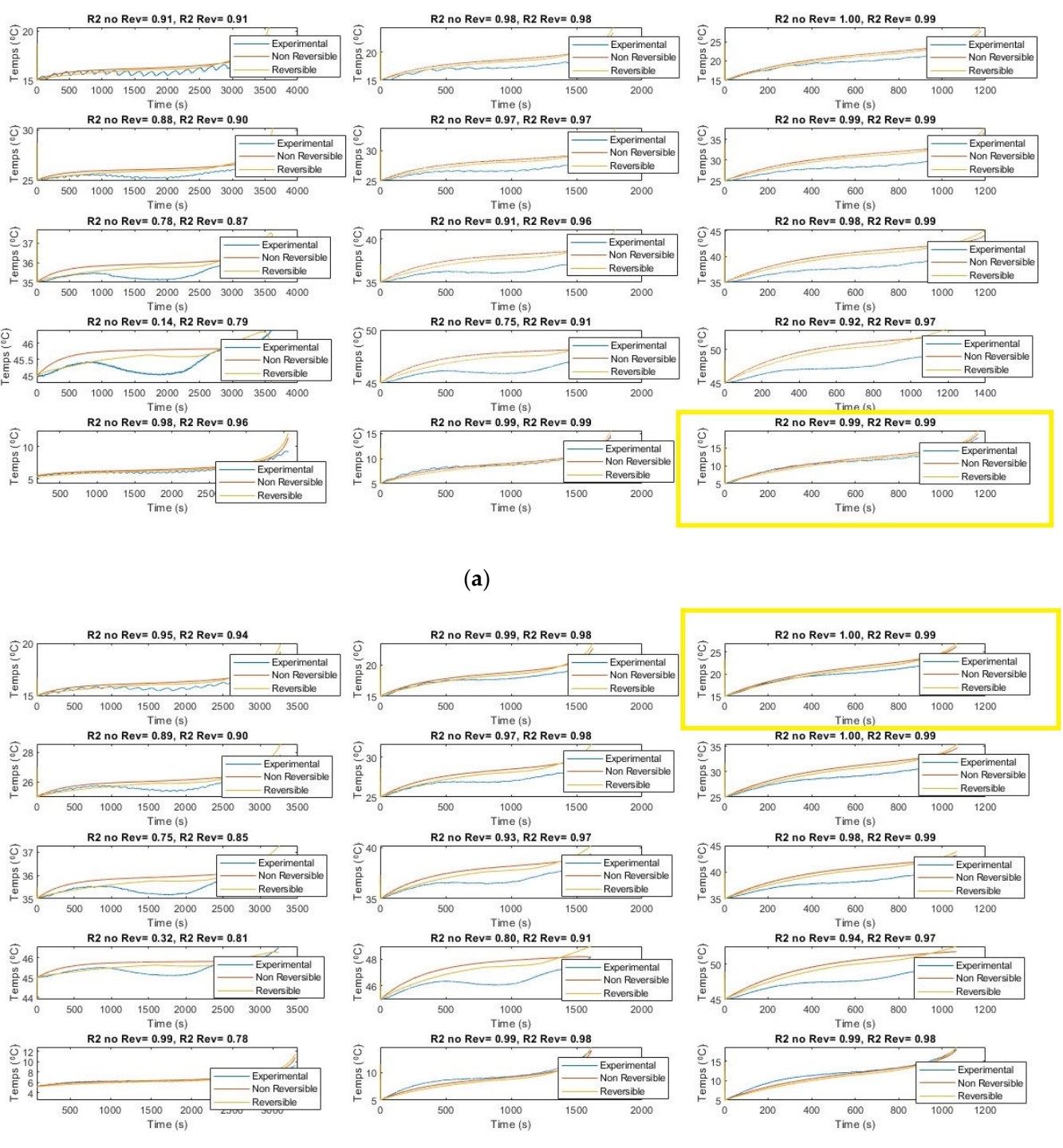

(**a**)

(**b**)

**Figure 5.** *Cont.*

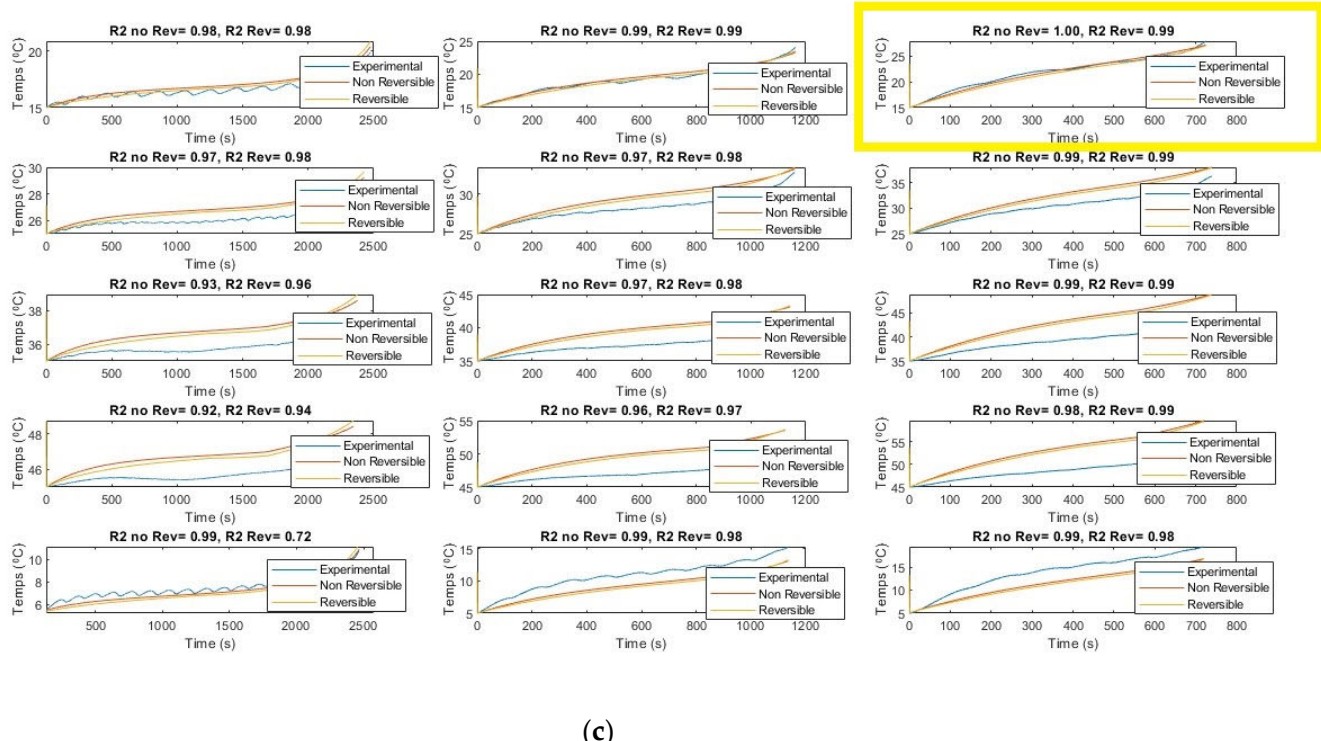

(**c**)

**Figure 5.** (**a**–**c**) Reversible and simple Rint heat models for battery at stated SOH at 5 different ambient temperatures and 3 stated C rates. (**a**) Simulated temperatures at ambient temperatures shown and with C rates of 1C, 2C, 3C from left to right with an SOH of 90%. (Highest Accuracy Reversible Model highlighted see Table 6 for details). (**b**) Simulated temperatures at ambient temperatures shown and with C rates of 1C, 2C, 3C from left to right with an SOH of 83% (Highest Accuracy Model highlighted). (**c**) Simulated temperatures at ambient temperatures shown and with C rates of 1C, 2C, 3C from left to right with an SOH of 65%. (Highest Accuracy Non-Reversible Model highlighted see Table 6 for details).

**Table 6.** Pearson Values for 0D thermal models with and without Reversible Heat Term. (Highest Accuracy Highlighted).

| SOH | Temp (K) | Current (Amps) | R No Reversible Heat | R Reversible Heat |
|---|---|---|---|---|
| 65% | 288 | 18 | 0.979 | 0.983 |
| 65% | 288 | 36 | 0.987 | 0.988 |
| 65% | 288 | 54 | 0.996 | 0.989 |
| 65% | T = 288 K ALL | | 0.992 | 0.986 |
| 65% | 298 | 18 | 0.968 | 0.982 |
| 65% | 298 | 36 | 0.973 | 0.982 |
| 65% | 298 | 54 | 0.992 | 0.99 |
| 65% | T = 298 K ALL | | 0.992 | 0.992 |
| 65% | 308 | 18 | 0.934 | 0.961 |
| 65% | 308 | 36 | 0.969 | 0.98 |
| 65% | 308 | 54 | 0.988 | 0.989 |
| 65% | T = 308 K ALL | | 0.989 | 0.992 |
| 65% | 318 | 18 | 0.915 | 0.945 |
| 65% | 318 | 36 | 0.963 | 0.974 |
| 65% | 318 | 54 | 0.984 | 0.985 |
| 65% | T = 318 K ALL | | 0.983 | 0.987 |
| 65% | 278 | 18 | 0.99 | 0.715 |
| 65% | 278 | 36 | 0.992 | 0.981 |
| 65% | 278 | 54 | 0.988 | 0.978 |

**Table 6.** *Cont.*

| SOH | Temp (K) | Current (Amps) | R No Reversible Heat | R Reversible Heat |
|---|---|---|---|---|
| 65% | T = 278 K ALL | | 0.984 | 0.916 |
| 65% | ALL T | | 0.997 | 0.997 |
| 83.40% | 288 | 18 | 0.951 | 0.945 |
| 83.40% | 288 | 36 | 0.991 | 0.976 |
| 83.40% | 288 | 54 | 0.997 | 0.986 |
| 83.40% | T = 288 K ALL | | 0.994 | 0.978 |
| 83.40% | 298 | 18 | 0.889 | 0.899 |
| 83.40% | 298 | 36 | 0.975 | 0.978 |
| 83.40% | 298 | 54 | 0.996 | 0.991 |
| 83.40% | T = 298 K ALL | | 0.993 | 0.986 |
| 83.40% | 308 | 18 | 0.751 | 0.85 |
| 83.40% | 308 | 36 | 0.927 | 0.966 |
| 83.40% | 308 | 54 | 0.977 | 0.989 |
| 83.40% | T = 308 K ALL | | 0.977 | 0.988 |
| 83.40% | 318 | 18 | 0.316 | 0.811 |
| 83.40% | 318 | 36 | 0.798 | 0.912 |
| 83.40% | 318 | 54 | 0.941 | 0.974 |
| 83.40% | T = 318 K ALL | | 0.937 | 0.972 |
| 83.40% | 278 | 18 | 0.987 | 0.777 |
| 83.40% | 278 | 36 | 0.992 | 0.985 |
| 83.40% | 278 | 54 | 0.993 | 0.984 |
| 83.40% | T = 278 K ALL | | 0.992 | 0.948 |
| 83.40% | ALL T | | 0.999 | 0.999 |
| 90.60% | 288 | 18 | 0.913 | 0.911 |
| 90.60% | 288 | 36 | 0.984 | 0.982 |
| 90.60% | 288 | 54 | 0.996 | 0.991 |
| 90.60% | T = 288 K ALL | | 0.99 | 0.979 |
| 90.60% | 298 | 18 | 0.879 | 0.895 |
| 90.60% | 298 | 36 | 0.971 | 0.972 |
| 90.60% | 298 | 54 | 0.993 | 0.99 |
| 90.60% | T = 298 K ALL | | 0.99 | 0.984 |
| 90.60% | 308 | 18 | 0.781 | 0.872 |
| 90.60% | 308 | 36 | 0.91 | 0.957 |
| 90.60% | 308 | 54 | 0.977 | 0.991 |
| 90.60% | T = 308 K ALL | | 0.971 | 0.987 |
| 90.60% | 318 | 18 | 0.136 | 0.791 |
| 90.60% | 318 | 36 | 0.745 | 0.91 |
| 90.60% | 318 | 54 | 0.921 | 0.973 |
| 90.60% | T = 318 K ALL | | 0.913 | 0.964 |
| 90.60% | 278 | 18 | 0.98 | 0.961 |
| 90.60% | 278 | 36 | 0.988 | 0.991 |
| 90.60% | 278 | 54 | 0.993 | 0.992 |
| 90.60% | T = 278 K ALL | | 0.992 | 0.983 |
| 90.60% | ALL T | | 0.998 | 0.998 |
| | T = 288 K | | 0.983 | 0.971 |
| | T = 298 K | | 0.983 | 0.975 |
| | T = 308 K | | 0.972 | 0.986 |
| | T = 318 K | | 0.922 | 0.965 |
| | T = 278 K | | 0.982 | 0.966 |

## 4. 2D Thermal Model

The 2D model solves the finite center difference in 2 dimensions as

$$
\begin{aligned}
T_n(i,j) = T_{n-1}(i,j) + dt * Dx * (T_{n-1}(i+1,j) - 2T_{n-1}(i,j) + T_{n-1}(i-1,j)) + dt * Dy \\
* (T_{n-1}(i,j+1) - 2T_{n-1}(i,j) + T_{n-1}(i,j-1)) + dt * \left( \dot{Q} \right)
\end{aligned}
\tag{6}
$$

where $T_n(i,j)$ is the current temperature at the current particular point in the mesh, $T_{n-1}(i,j)$ is the temperature at the current point in the mesh at the previous time step, $T_n(i+1,j)$ is the temperature at the current time one step forward in the x direction and $T_n(i,j+1)$ is the temperature at the current time one step forward in the y direction.

$\dot{Q}$ is the sum of the drains and sources of heat generation, namely the heat of reaction, the heat due to internal resistance and the convection loss

$$\dot{Q} = dt * \left( \frac{hA_s * (T_n(i,j) - T_{amb}) + I^2 R_{int} + IT\frac{dV_{ocv}}{dT}}{mc_p} \right) \tag{7}$$

where $Dx$ and $Dy$ are related to the thermal diffusivity $D$ as:

$$D = \frac{K}{\rho c_p} \tag{8}$$

$$Dx = \frac{D}{dx^2} \tag{9}$$

$$Dy = \frac{D}{dy^2} \tag{10}$$

These set of equations are integrated through time using Euler's method ensuring that the time step is well below the Lyapunov Stability condition neglecting the heat source. The condition is show as:

$$dt \leq \frac{1}{2D\left(\frac{1}{dx^2} + \frac{1}{dy^2}\right)} \tag{11}$$

The mesh is made by dividing the rectangular battery in 32 sections lengthwise and widthwise. The Boundary and initial conditions are shown below:

$$T_0(i,j) = T_{amb} \tag{12}$$

$$T_{OuterEdges} = T_{amb} + dt * \left( \frac{hA_s * (T_n(i,j) - T_{amb})}{mc_p} \right) \tag{13}$$

$$T_{Tab} = T_{amb} + dt * \left( \frac{I^2 R_{tab} + hA_s * (T_n(i,j) - T_{amb})}{mc_p} \right) \tag{14}$$

The parameter values for the simulation of the three different batteries in the experiments at three different SOHs are provided in Table 7.

**Table 7.** 2D Thermal Model Parameter Values for Battery at 3 Experimental SOHs 65%, 85% and 91% in order.

| Parameter | Value |
|---|---|
| $K_1$ | 0.038 W/m-K |
| $K_2$ | 0.018 W/m-K |
| $K_3$ | 0.062 W/m-K |
| $C_{p1}$ | 3986 J/Kg-K |
| $C_{p2}$ | 2987 J/Kg-K |
| $C_{p3}$ | 2021 J/Kg-K |
| $\rho_1$ | 5302 Kg/m$^3$ |
| $\rho_2$ | 5288 Kg/m$^3$ |
| $\rho_3$ | 5001 Kg/m$^3$ |

**Table 7.** *Cont.*

| Parameter | Value |
|:---:|:---:|
| $h_1$ | 32 W/m$^2$-K |
| $h_2$ | 32 W/m$^2$-K |
| $h_3$ | 43 W/m$^2$-K |
| $R_{TabCathode1}$ | 0.177 ohms |
| $R_{TabCathode2}$ | 0.164 ohms |
| $R_{TabCathode3}$ | 0.185 ohms |
| $R_{TabAnode1}$ | 0.175 ohms |
| $R_{TabAnode2}$ | 0.175 ohms |
| $R_{TabAnode3}$ | 0.155 ohms |

We see the specific heat capacity and the density increase with degradation.

Figure 6 shows the simulated output from the finite difference simulation. We note the boundary conditions with the tab heat source is included in the simulation. The surface mesh is seen to be divided into 32 sections both lengthwise and widthwise. Each battery at each individual SOH has its own simulation parameters.

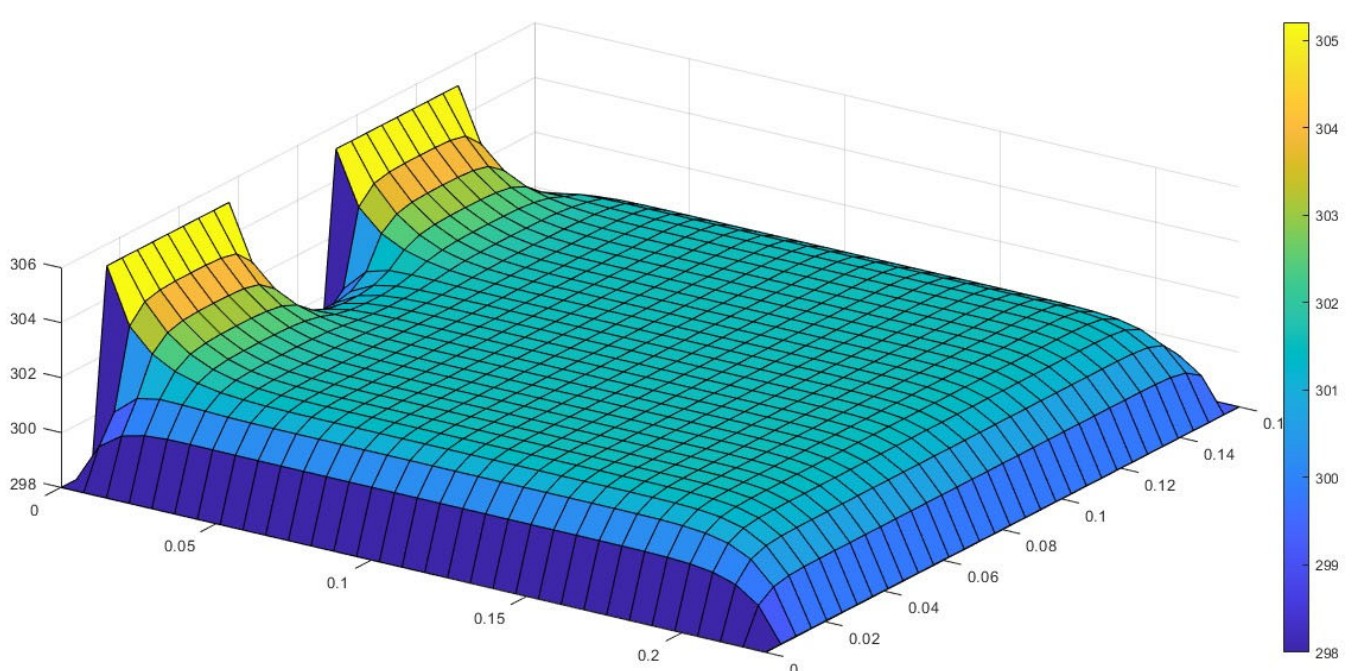

**Figure 6.** Sample of Surface Temperature (K) Produced by 2D Model.

Figure 7 shows the beginning of discharge, middle of discharge and end of discharge time steps for all 45 experimental runs. The three C rates are presented in each row, while the time of the discharge is presented in the columns.

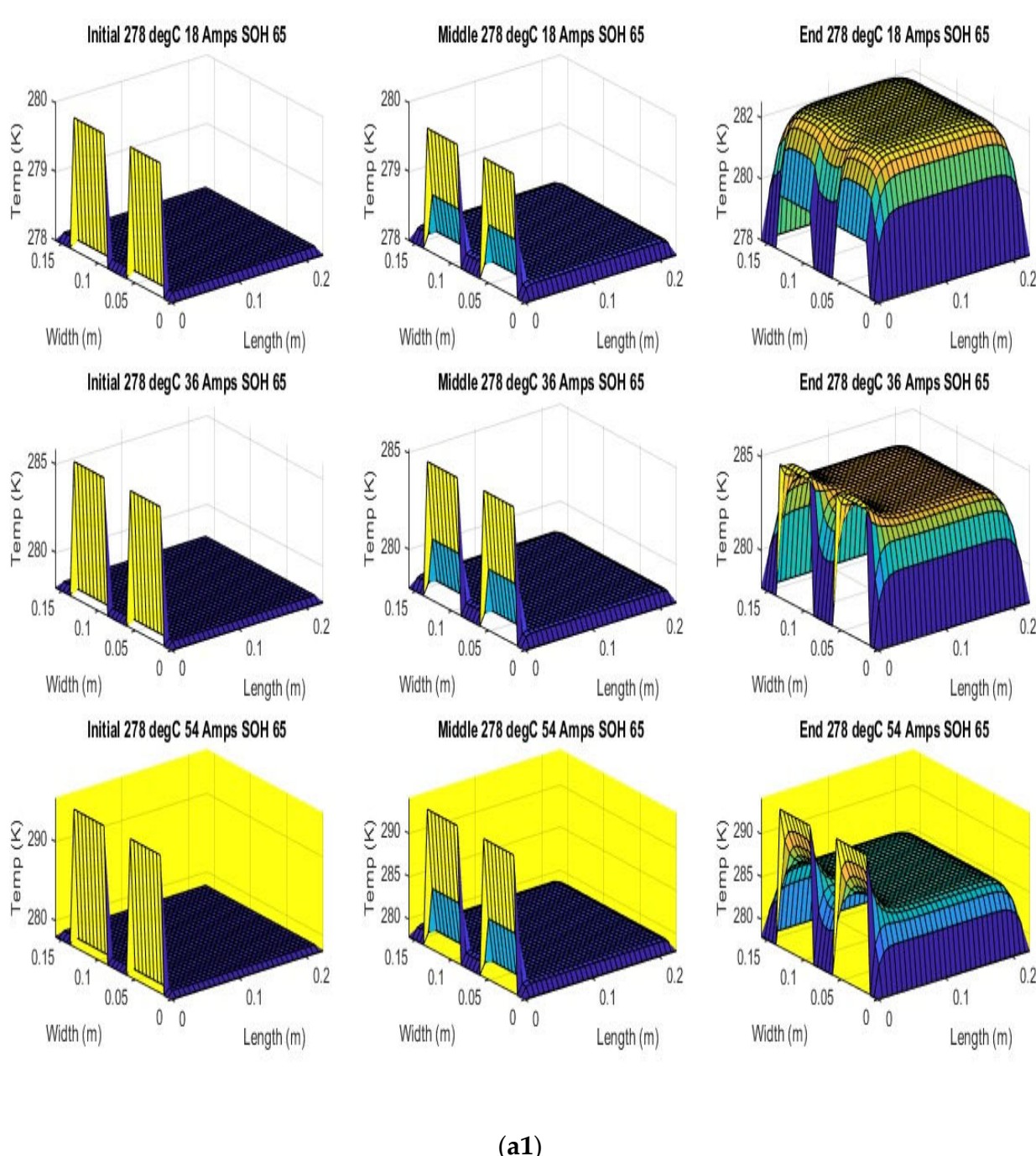

(**a1**)

**Figure 7.** *Cont.*

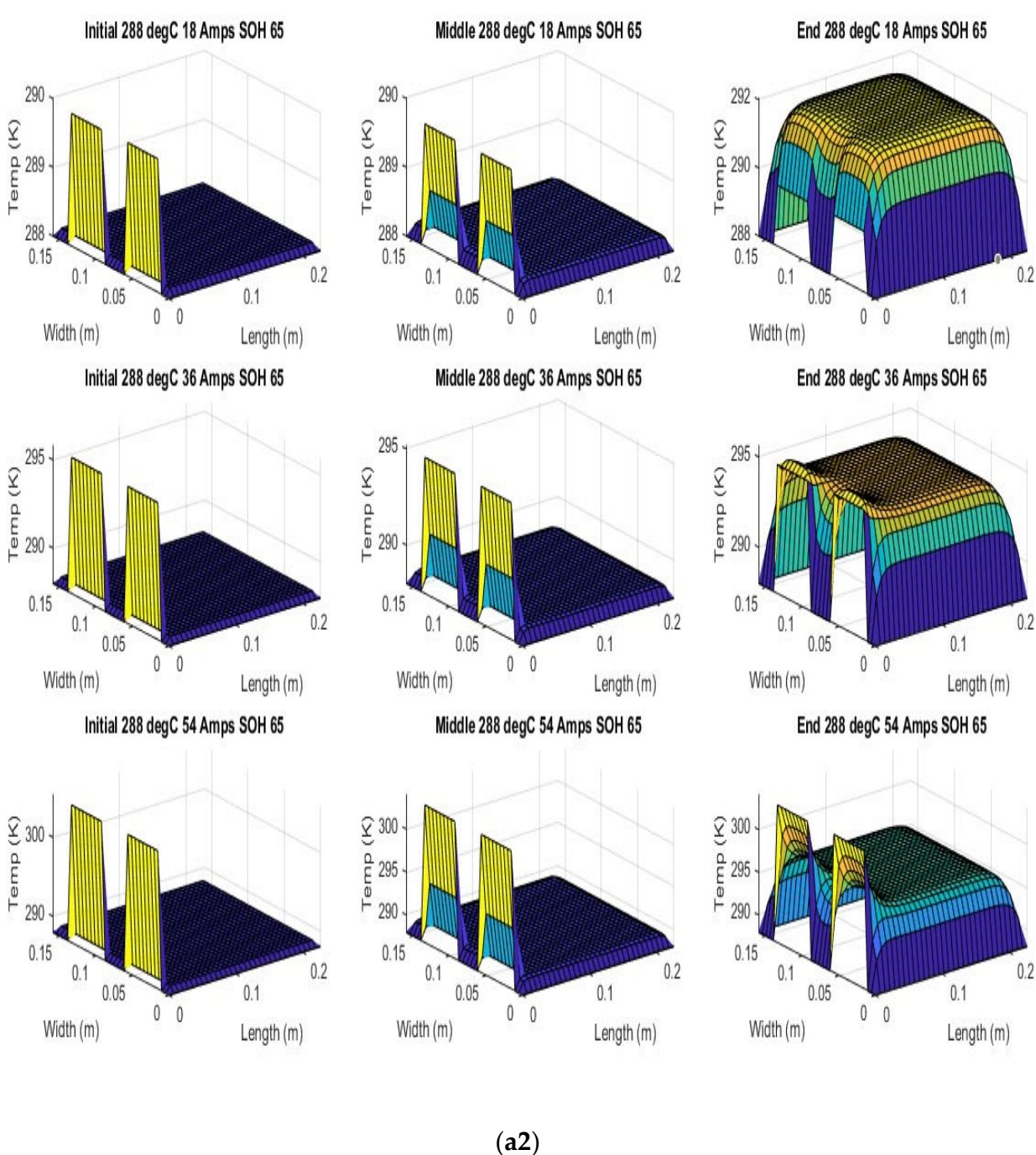

(**a2**)

**Figure 7.** *Cont.*

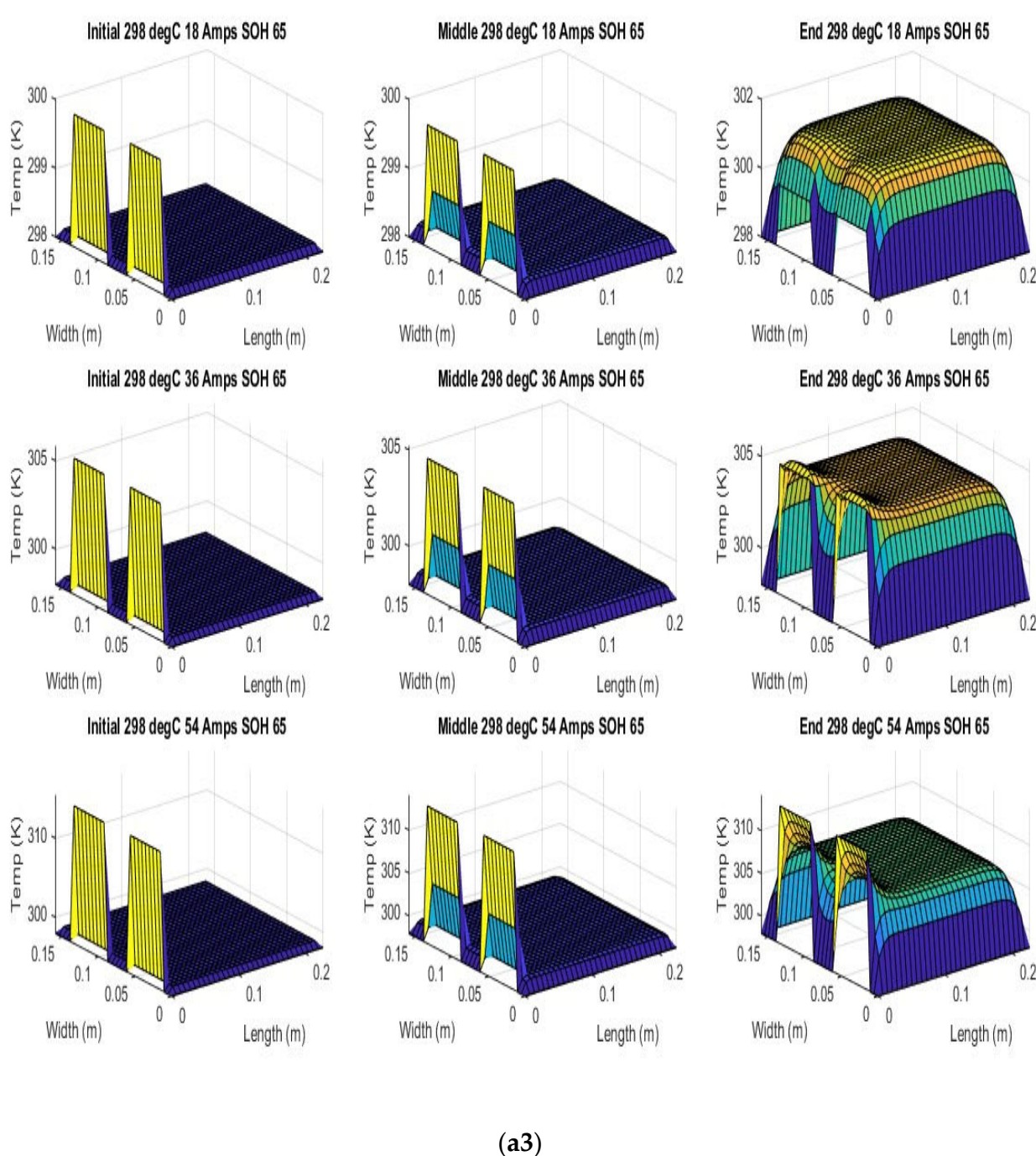

(**a3**)

**Figure 7.** *Cont*.

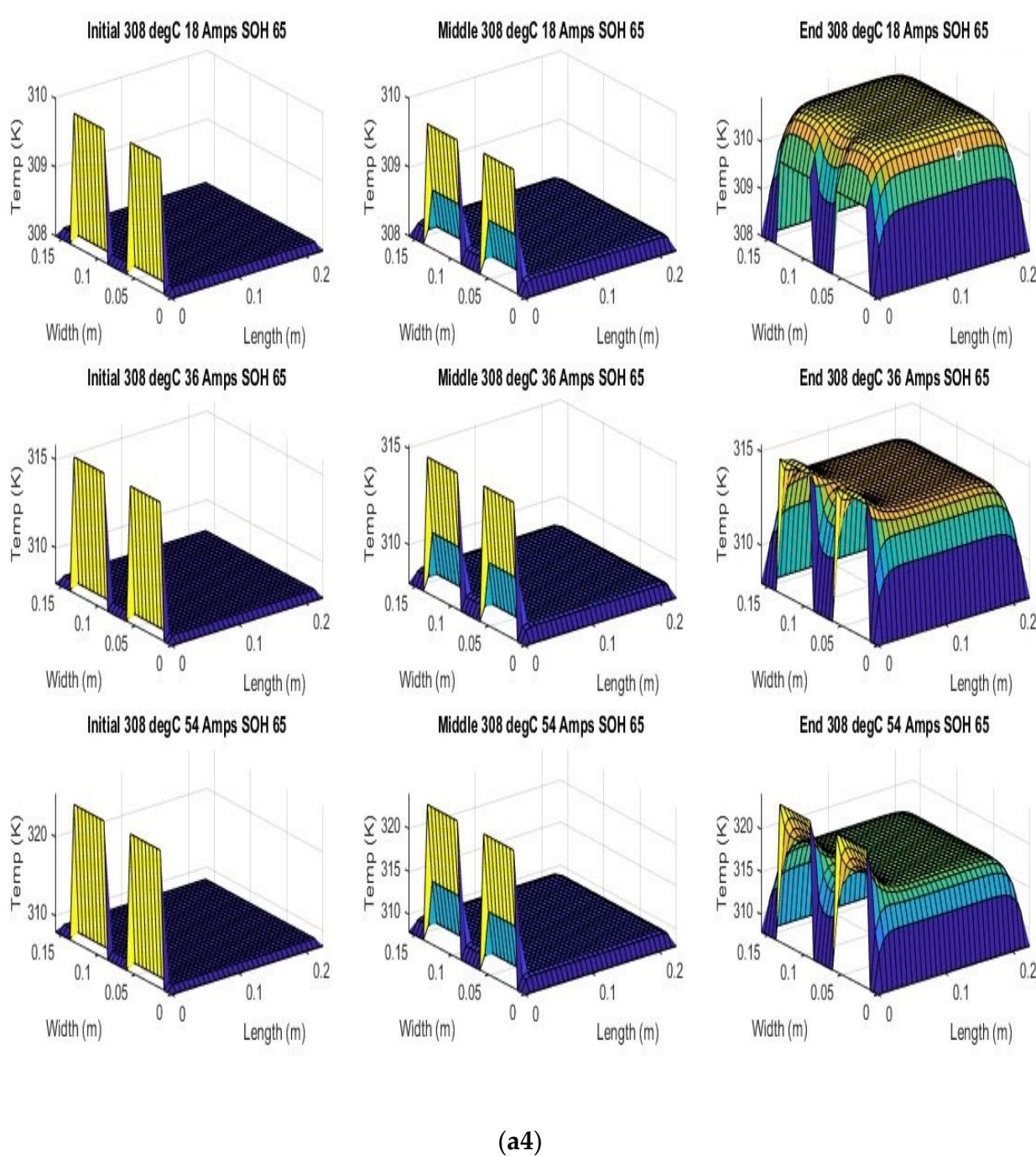

(**a4**)

**Figure 7.** *Cont.*

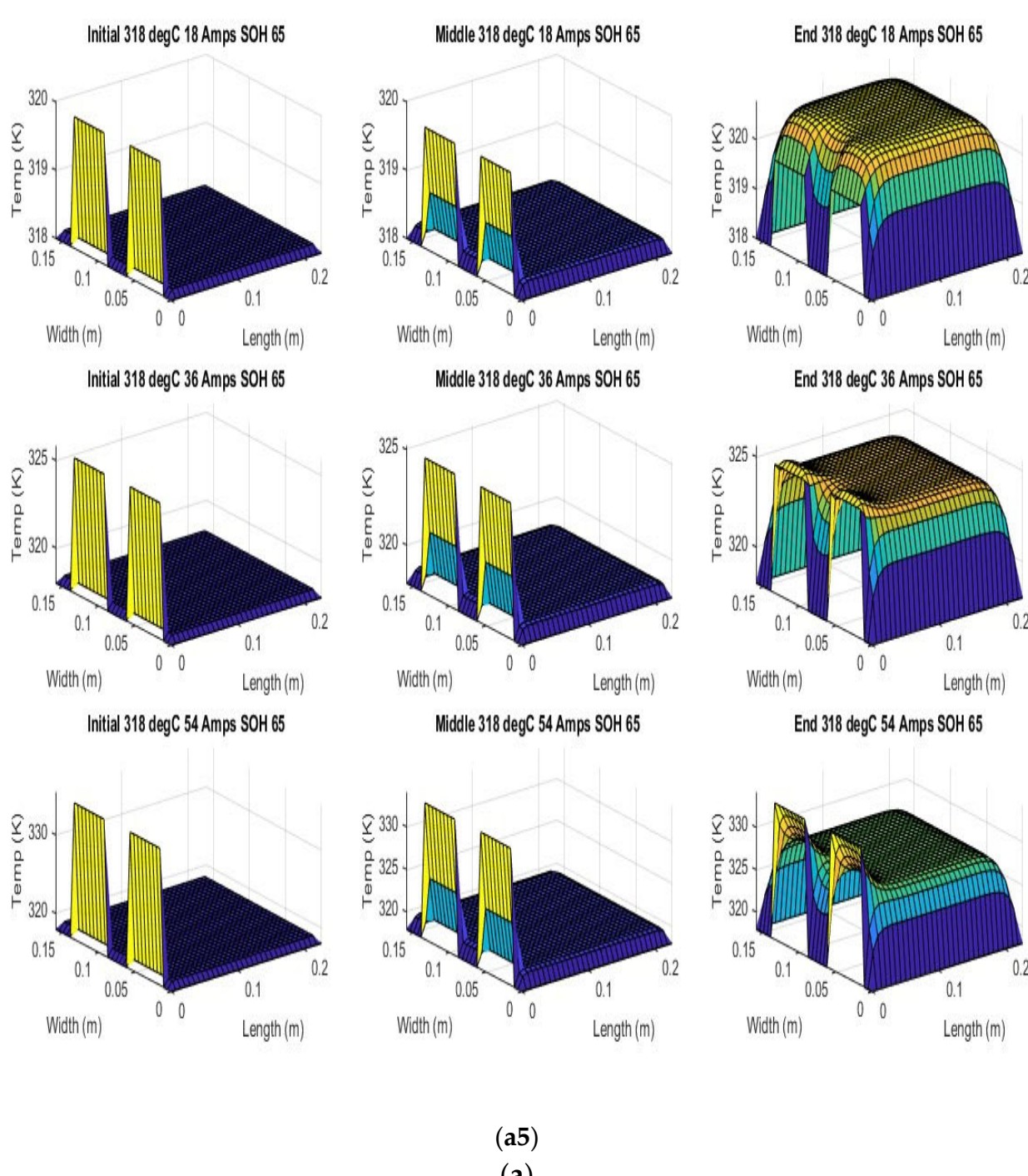

(**a5**)

(**a**)

**Figure 7.** *Cont.*

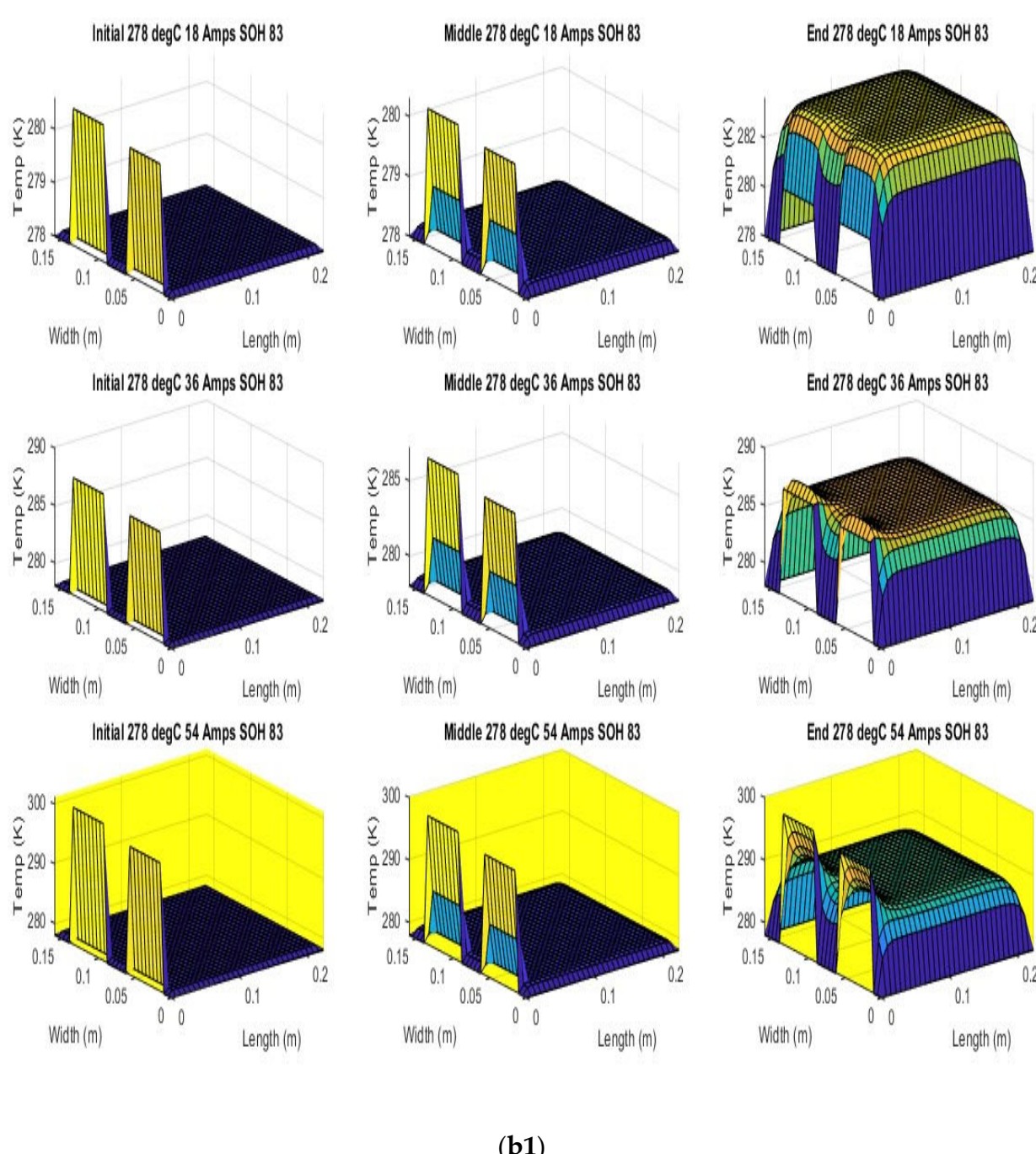

(**b1**)

**Figure 7.** *Cont.*

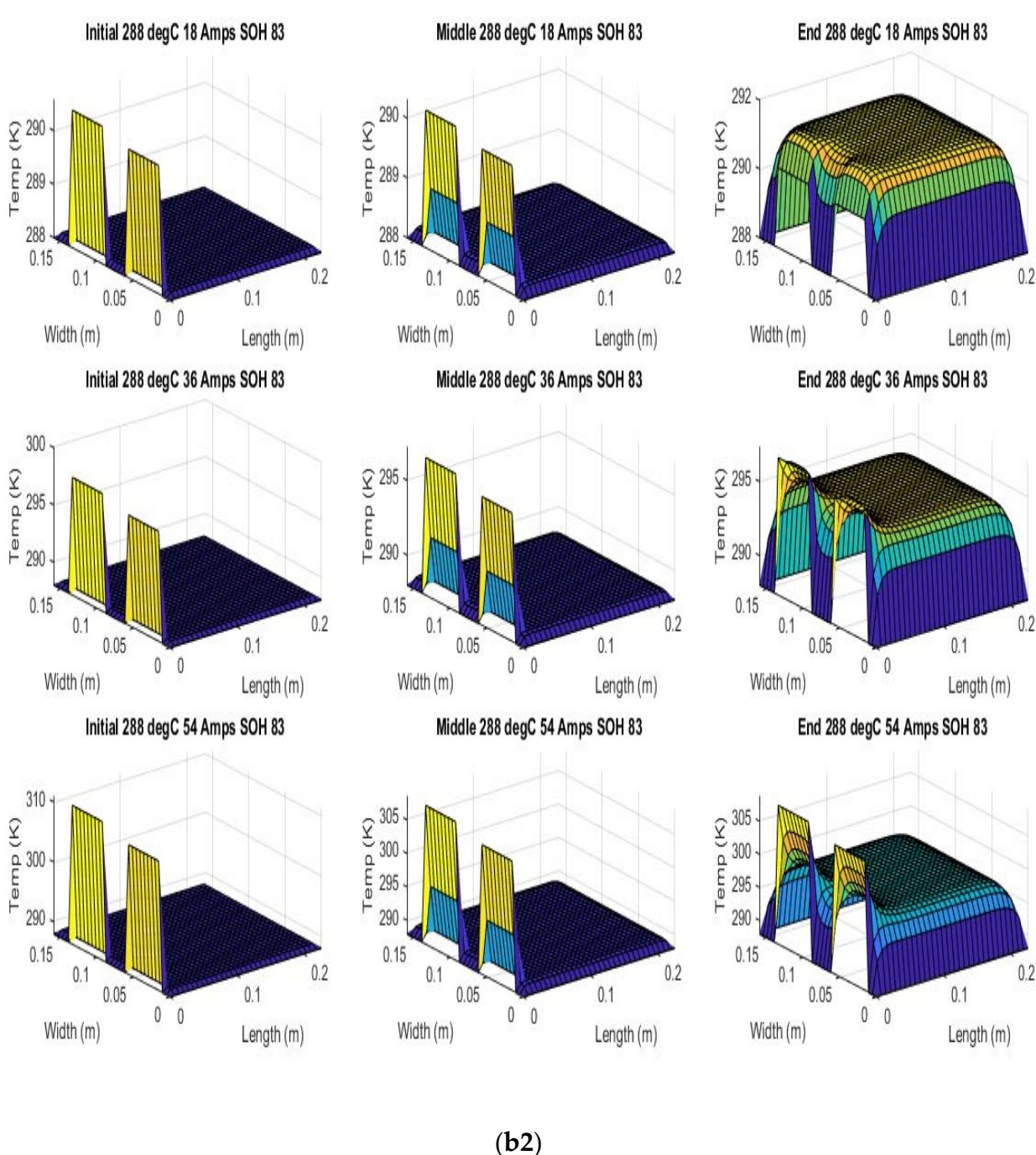

(**b2**)

**Figure 7.** *Cont.*

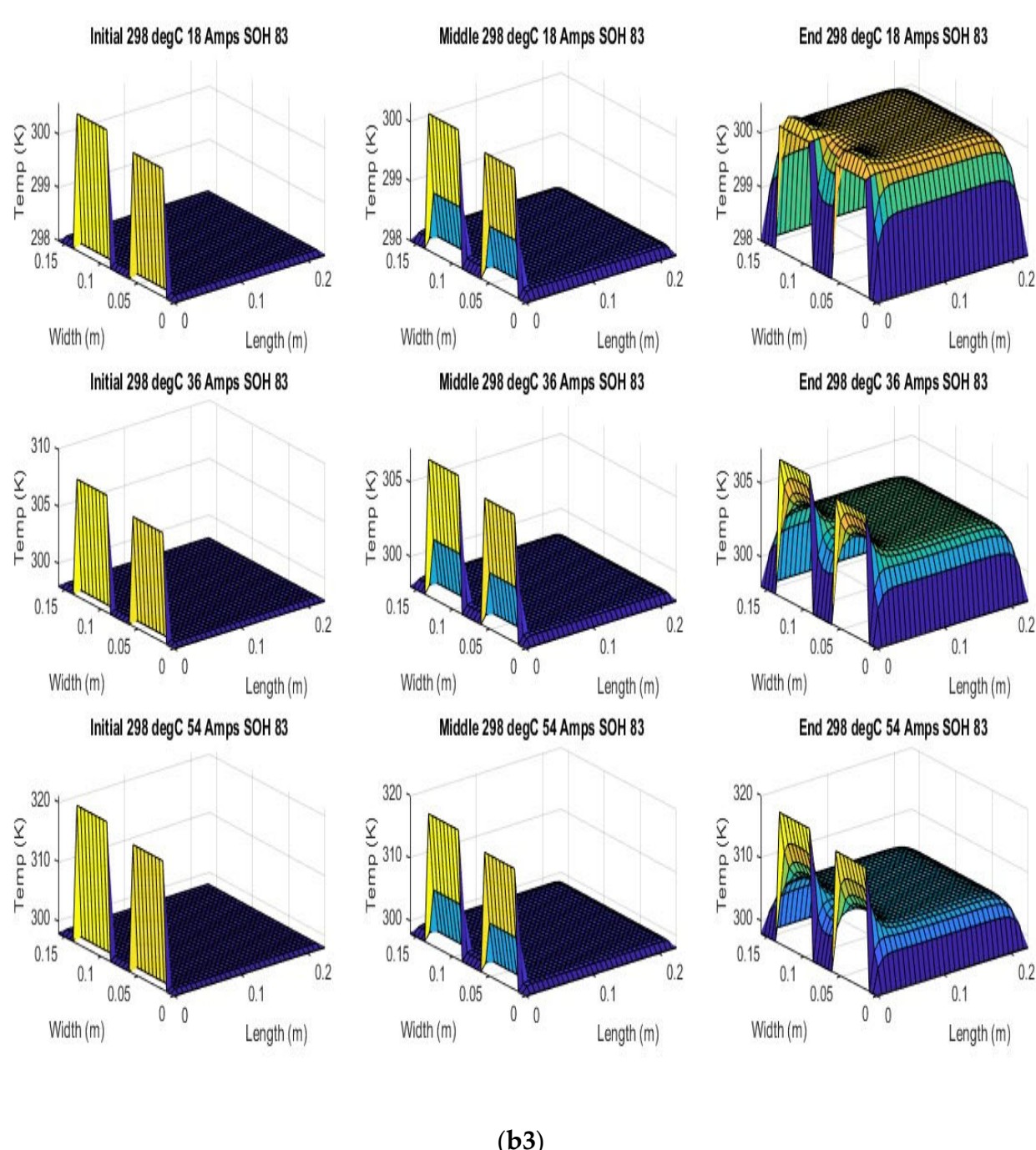

(**b3**)

**Figure 7.** *Cont.*

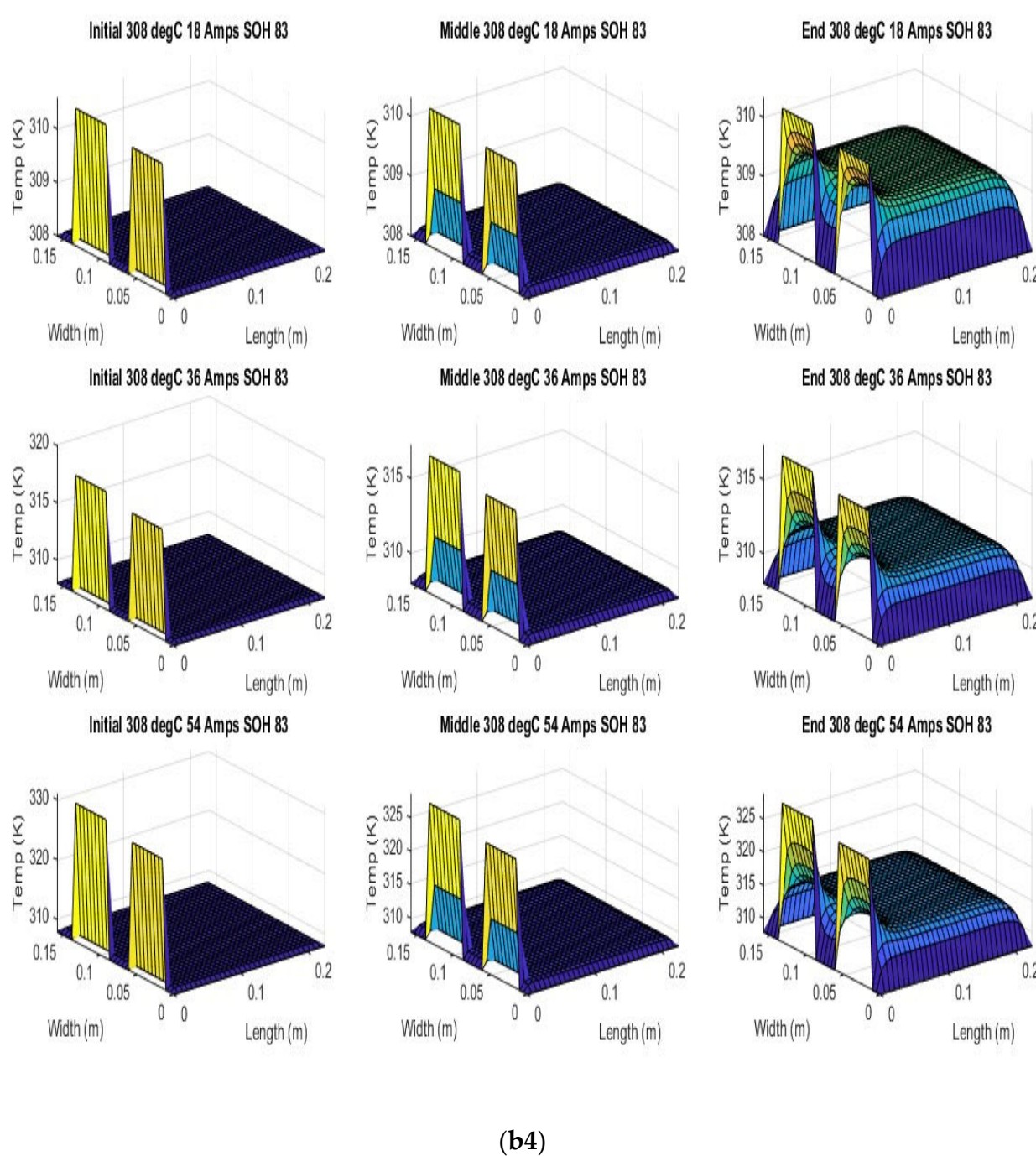

(**b4**)

**Figure 7.** *Cont.*

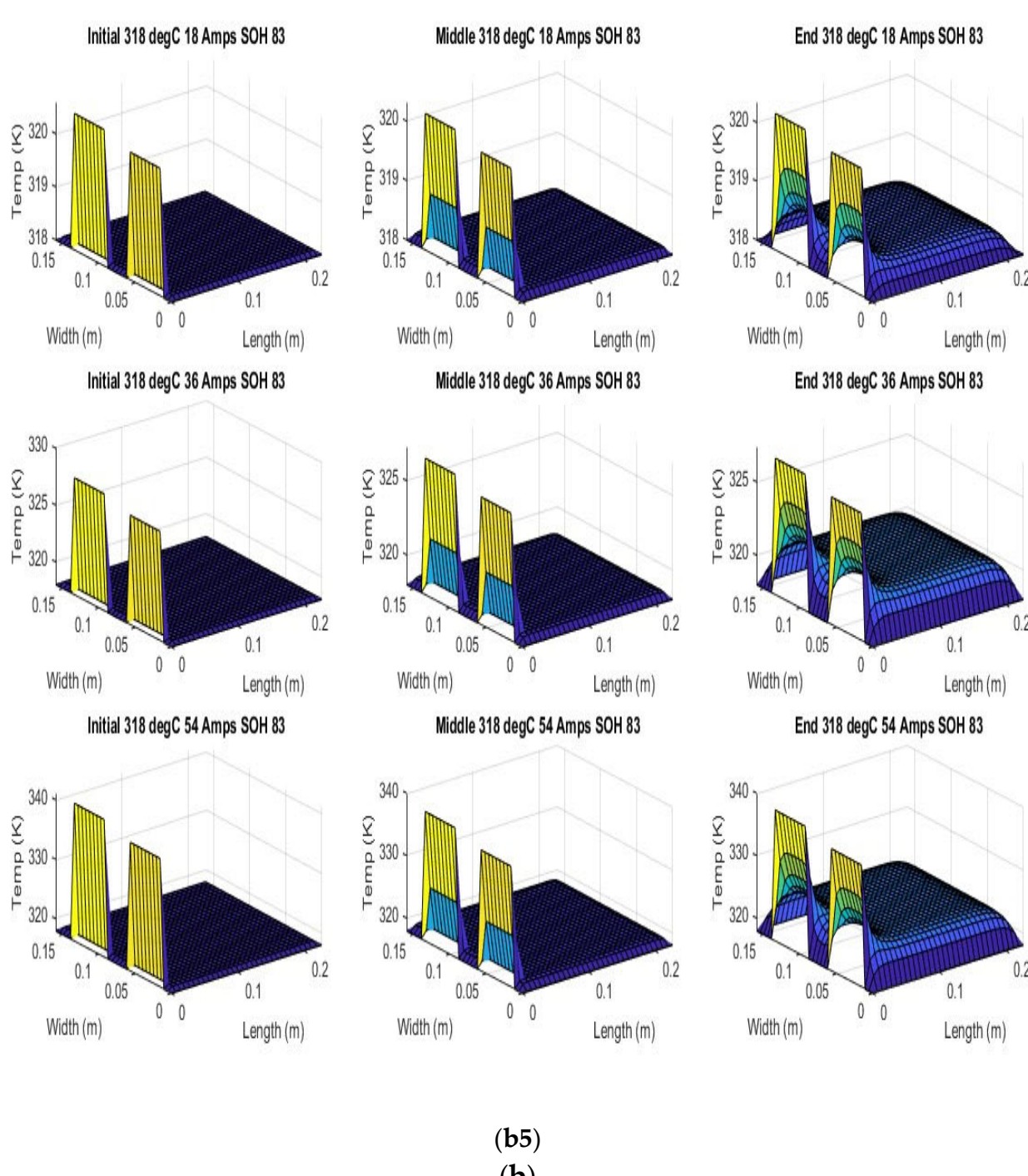

(**b5**)

(**b**)

**Figure 7.** *Cont.*

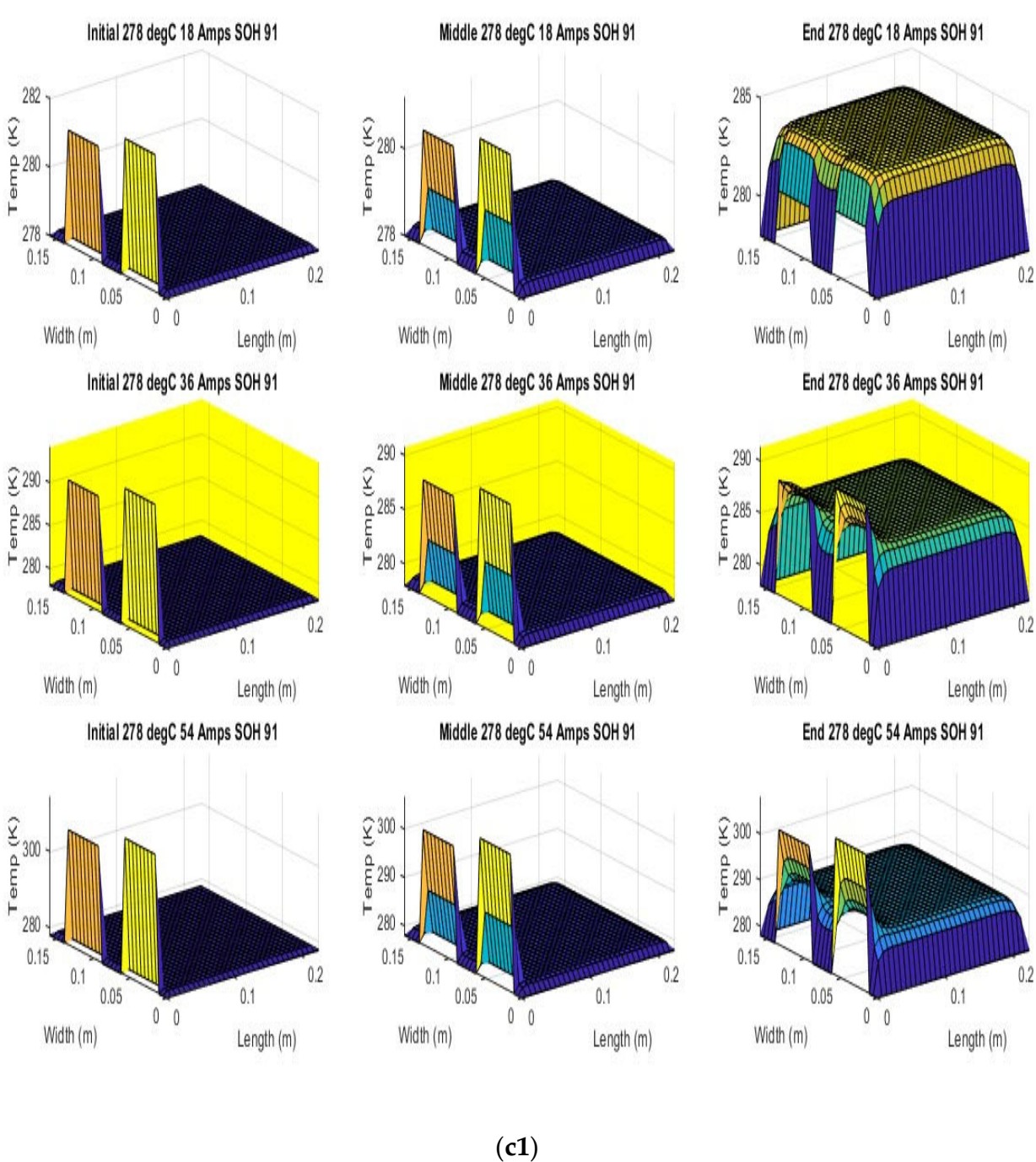

(**c1**)

**Figure 7.** *Cont.*

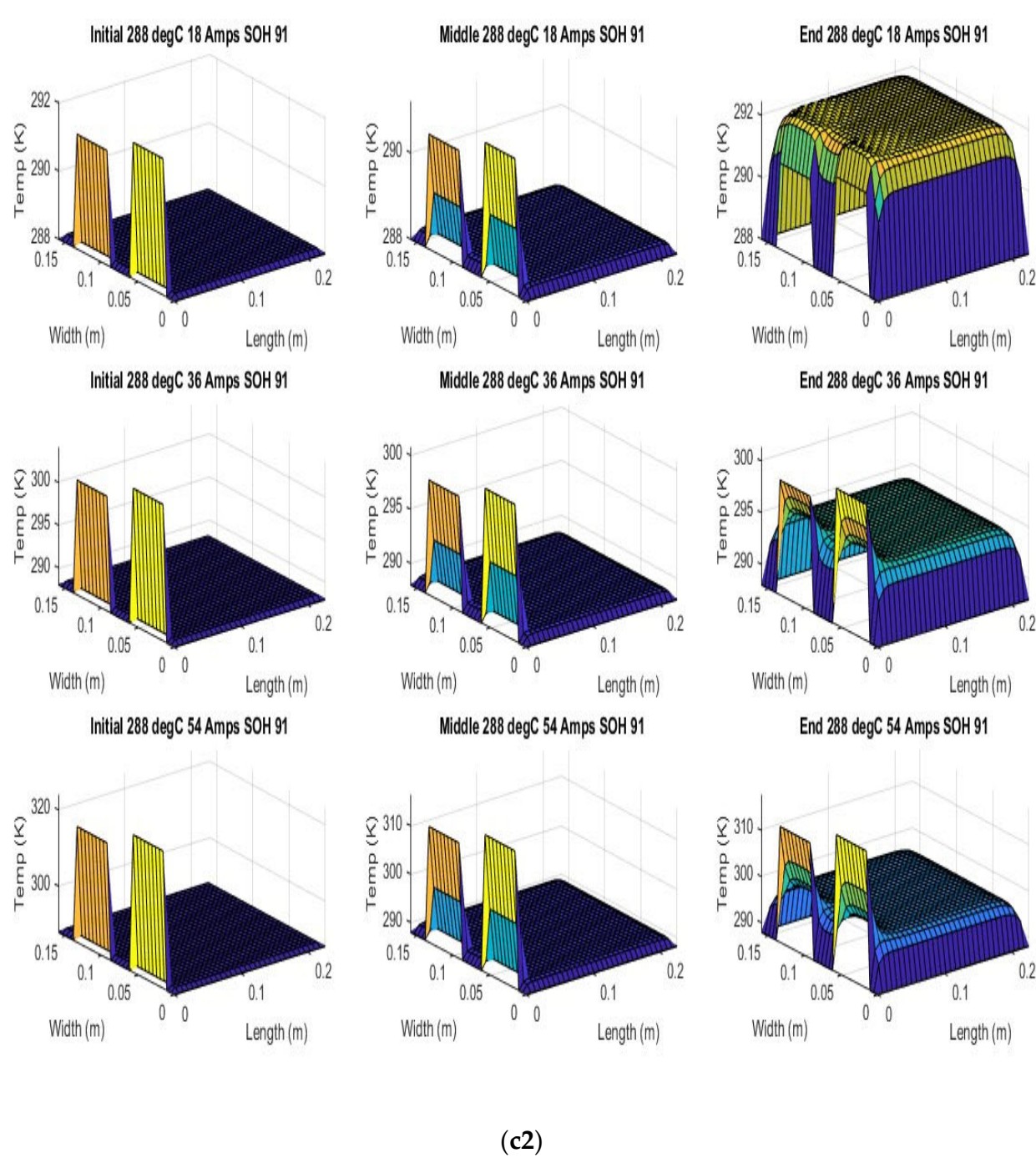

(**c2**)

**Figure 7.** *Cont.*

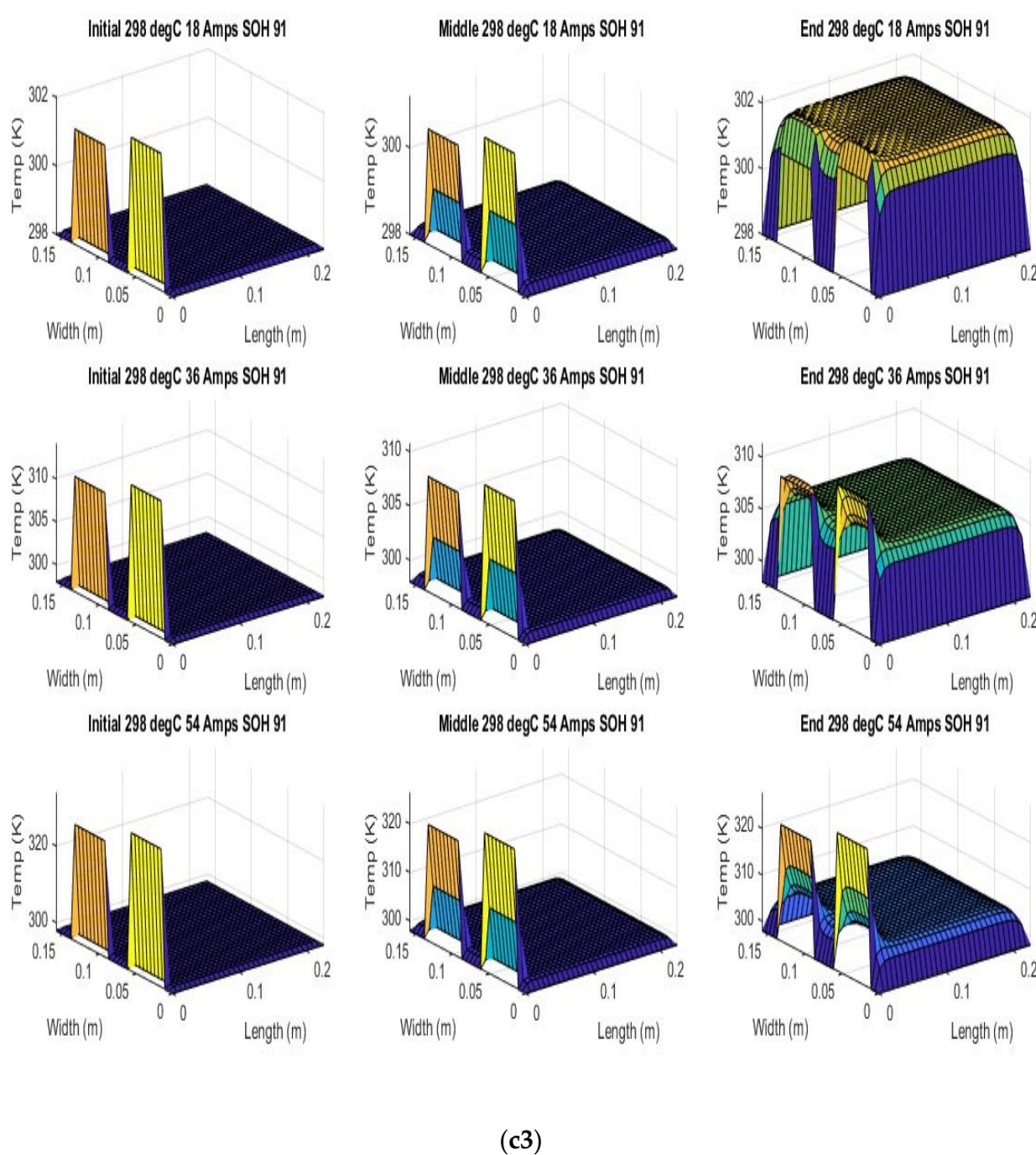

(**c3**)

**Figure 7.** *Cont.*

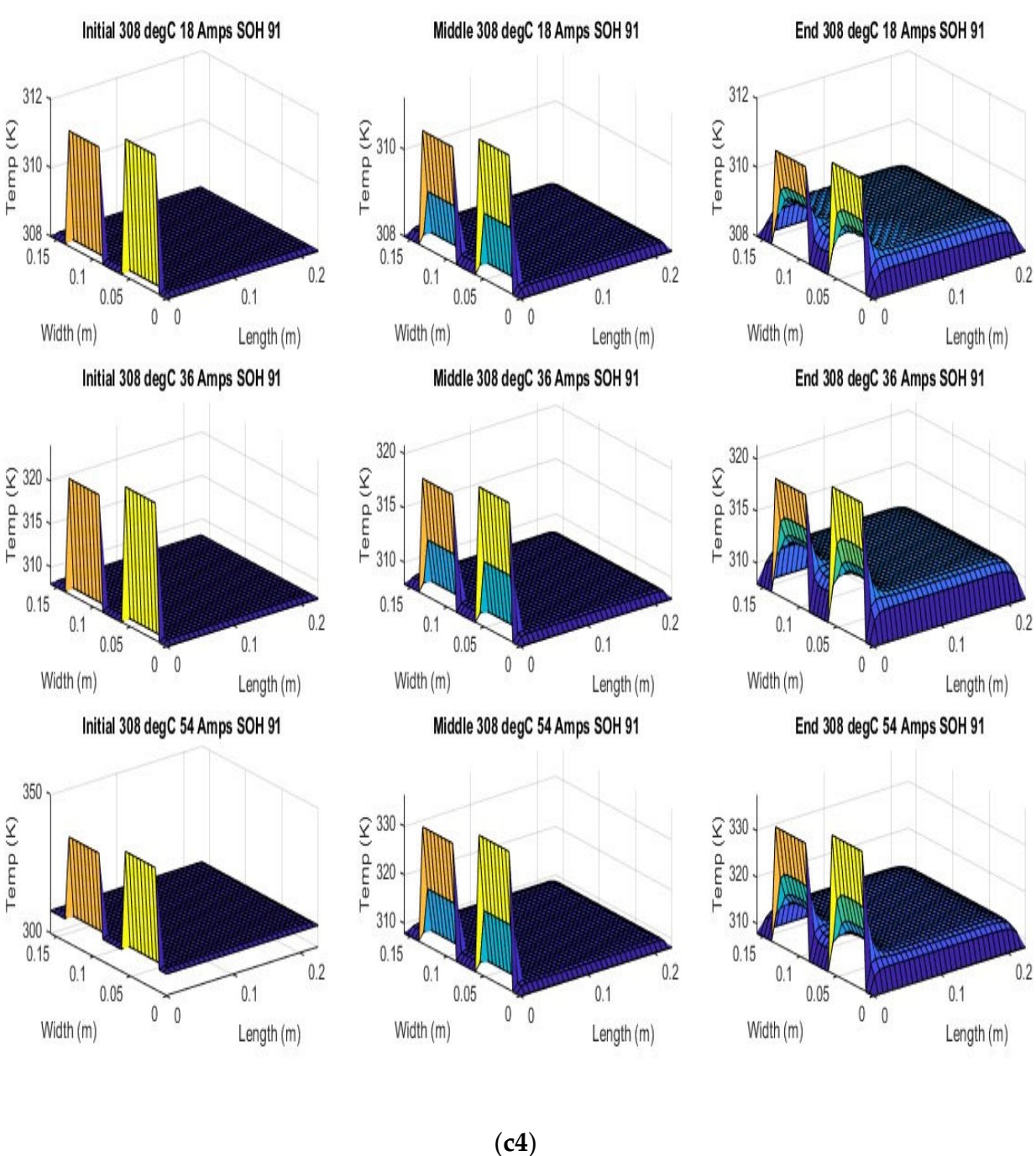

**(c4)**

**Figure 7.** *Cont.*

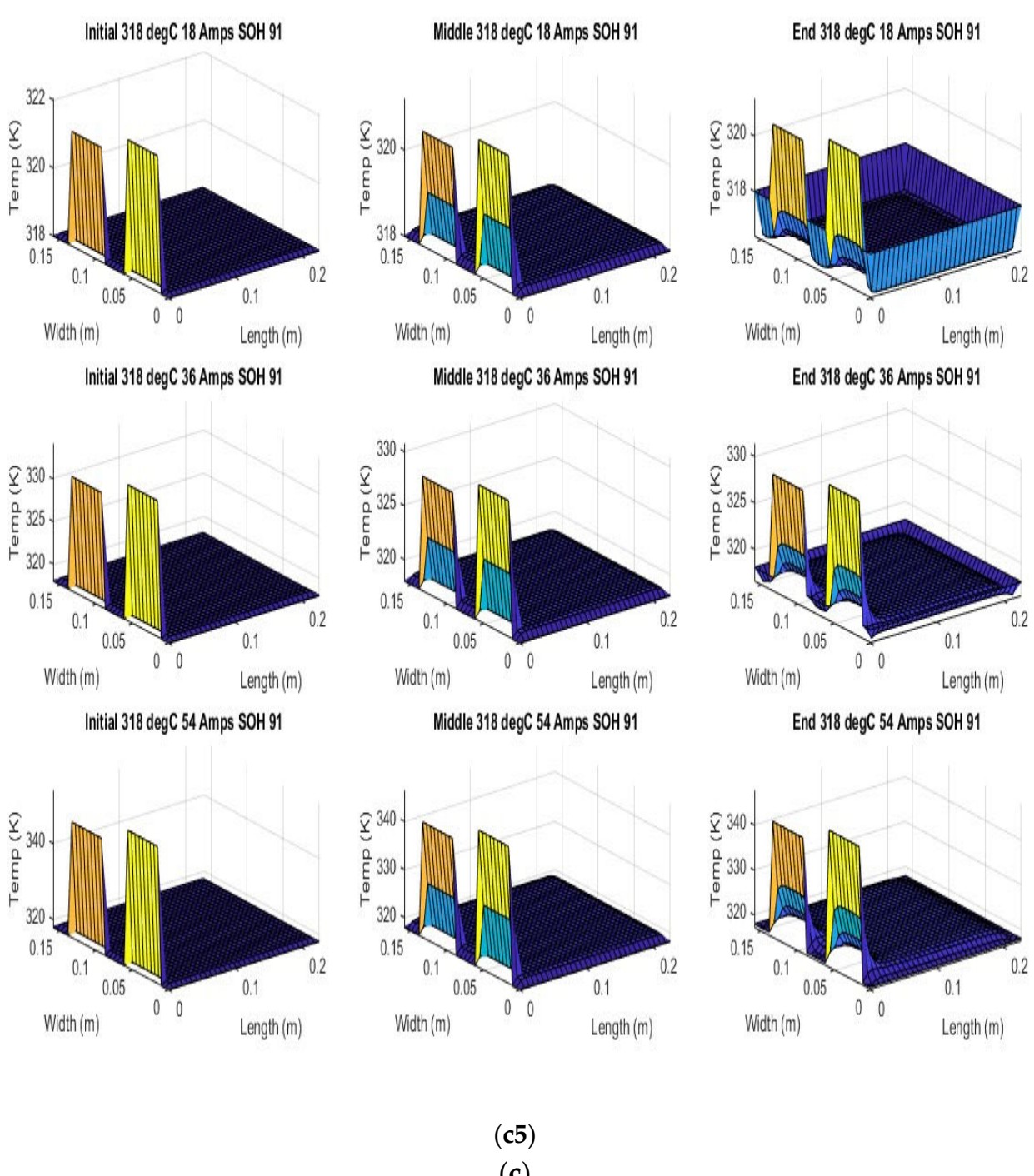

(c5)

(c)

**Figure 7.** Surface Temperatures (K) Produced by 2D model for all 45 Experimental Temperatures, SOHs and C rates. (**a**) Initial, middle and end Surface Temperatures for SOH 65% at 278 K, 288 K, 298 K, 308 K, 318 K for 1C, 2C, 3C C-rates (Highest Accuracy Model at SOH 65% highlighted). (**b**) Initial, middle and end Surface Temperatures for SOH 83% at 278 K, 288 K, 298 K, 308 K, and 318 K Initial, for 1C, 2C, and 3C C-rates (Highest Accuracy Model at SOH 83% highlighted). (**c**) Initial, middle and end Surface Temperatures for SOH 91% at 278 K, 288 K, 298 K, 308 K, and 318 K for 1C, 2C, and 3C C-rates (Highest Accuracy of all 45 Runs Highlighted).

The models show good agreement with experimental results and are generally numerically stable for all conditions showing that the model is robust and the timestep is adequately small and satisfies the stability conditions of all experimental runs even given the steep temperature gradient between the tabs and the battery surface.

The Pearson coefficients for the 2D model are found in Table 8. They are found to be in close agreement with experimental results using only the thermal parameters found in Table 7.

**Table 8.** Pearson Coefficients for 2D Model at Given SOHS, Ambient Temperatures and Currents. (Highest Accuracies Highlighted).

| SOH | $T_{amb}$ | Current | R |
|---|---|---|---|
| 65% | 288 | 18 | 0.881 |
| 65% | 288 | 36 | 0.916 |
| 65% | 288 | 54 | 0.952 |
| 65% | T = 288 Total | | 0.962 |
| 65% | 298 | 18 | 0.843 |
| 65% | 298 | 36 | 0.889 |
| 65% | 298 | 54 | 0.929 |
| 65% | T = 298 Total | | 0.956 |
| 65% | 308 | 18 | 0.797 |
| 65% | 308 | 36 | 0.871 |
| 65% | 308 | 54 | 0.911 |
| 65% | T = 308 Total | | 0.944 |
| 65% | 318 | 18 | 0.774 |
| 65% | 318 | 36 | 0.855 |
| 65% | 318 | 54 | 0.881 |
| 65% | T = 318 Total | | 0.926 |
| 65% | 278 | 18 | 0.918 |
| 65% | 278 | 36 | 0.947 |
| 65% | 278 | 54 | 0.956 |
| 65% | T = 278 Total | | 0.962 |
| 65% | All T | | 0.996 |
| 83.4% | 288 | 18 | 0.912 |
| 83.4% | 288 | 36 | 0.955 |
| 83.4% | 288 | 54 | 0.952 |
| 83.4% | T = 288 Total | | 0.98 |
| 83.4% | 298 | 18 | 0.83 |
| 83.4% | 298 | 36 | 0.922 |
| 83.4% | 298 | 54 | 0.918 |
| 83.4% | T = 298 Total | | 0.96 |
| 83.4% | 308 | 18 | 0.741 |
| 83.4% | 308 | 36 | 0.855 |
| 83.4% | 308 | 54 | 0.9 |
| 83.4% | T = 308 Total | | 0.933 |
| 83.4% | 318 | 18 | 0.395 |
| 83.4% | 318 | 36 | 0.788 |
| 83.4% | 318 | 54 | 0.866 |

**Table 8.** *Cont.*

| SOH | $T_{amb}$ | Current | R |
|---|---|---|---|
| 83.4% | T = 318 Total | | 0.891 |
| 83.4% | 278 | 18 | 0.951 |
| 83.4% | 278 | 36 | 0.972 |
| 83.4% | 278 | 54 | 0.973 |
| 83.4% | T = 278 Total | | 0.984 |
| 83.4% | ALL T | | 0.999 |
| 91% | 288 | 18 | 0.867 |
| 91% | 288 | 36 | 0.945 |
| 91% | 288 | 54 | 0.936 |
| 91% | T = 288 Total | | 0.968 |
| 91% | 298 | 18 | 0.82 |
| 91% | 298 | 36 | 0.845 |
| 91% | 298 | 54 | 0.881 |
| 91% | T = 298 Total | | 0.939 |
| 91% | 308 | 18 | 0.777 |
| 91% | 308 | 36 | 0.809 |
| 91% | 308 | 54 | 0.764 |
| 91% | T = 308 Total | | 0.887 |
| 91% | 318 | 18 | −0.297 |
| 91% | 318 | 36 | 0.299 |
| 91% | 318 | 54 | 0.562 |
| 91% | T = 318 Total | | 0.773 |
| 91% | 278 | 18 | 0.961 |
| 91% | 278 | 36 | 0.978 |
| 91% | 278 | 54 | 0.976 |
| 91% | T = 278 Total | | 0.982 |
| 91% | All T | | 0.999 |
| | T = 288 K | | 0.957 |
| | T = 298 K | | 0.934 |
| | T = 308 K | | 0.898 |
| | T = 318 K | | 0.805 |
| | T = 278 K | | 0.965 |

The table shows the model is more accurate at low temperatures than at high temperatures. The model is most accurate at 83% SOH and least accurate at 65% SOH with 91% SOH being in between. The highest accuracy single test is at 91% SOH, 278 K and 36 Amps. The highest accuracy models for SOH 65% and 83% are at 278 K and 54 Amps whereas for 91% there is a deviation from trend occurring at 36 Amps and still at 278 K which is our experimentally lowest temperature. The overall $R^2$ value for the 2D thermal model was 0.996.

## 5. Conclusions

The purpose of the paper was to create models that utilize practically measurable parameters to simulate a lithium ion battery's internal resistance and surface temperature. Overall, we see that all models have a high accuracy with the data. Equation (3) shows the internal resistance as the sum of five terms, where the first term is a simple constant. We see a term with asinh of both the current and SOC expected to be related to the activation overpotential and a term with an exponential of the reciprocal of temperature expected to be related to mass transfer losses. The term is multiplied by the $A_9$ parameter, which is a function of the SOC being zero at SOCs above 35%, contributing little to the internal resistance.

The internal resistance model is then used in three thermal models: two 0D models and one 2D Finite Element model. The 0D models are with and without reversible heat terms. The reversible heat is calculated using the Bernardi heat equation. The 2D model also adds tab heating junction resistance as heat source terms for each table. We see that for both the 2D and 0D models, the simulations are more accurate at lower temperatures. We see that the 0D reversible heat model is most accurate at the lowest temperature of 278 K, while the non-reversible model is most accurate at 288 K. We see the 0D model's most accurate runs are at the highest current of 54 Amps, while the 2D model's most accurate run is at 36 Amps. We note that Table 7 of the 2D model shows the specific heat capacity and the density increase with degradation and increases in SOH. We see that the 0D non-reversible model was most accurate with the 0D reversible model coming in second and the 2D model coming in third with $R^2$ values of 0.9964, 0.9962 and 0.996, respectively.

**Author Contributions:** Conceptualization, A.M., S.P. and M.F.; Data curation, A.M. and Y.S.; Formal analysis, A.M., M.K.T. and M.F.; Funding acquisition, S.P., M.F. and R.F.; Investigation, A.M., M.K.T. and S.P.; Methodology, A.M.; Project administration, Y.S., S.P., M.F. and R.F.; Resources, M.F. and R.F.; Software, A.M. and M.K.T.; Supervision, A.M., S.P., M.F. and R.F.; Validation, Y.S. and M.K.T.; Writing—original draft, A.M.; Writing—review & editing, Y.S. All authors have read and agreed to the published version of the manuscript.

**Funding:** This research received no external funding.

**Institutional Review Board Statement:** Not applicable.

**Informed Consent Statement:** Not applicable.

**Data Availability Statement:** Not applicable.

**Conflicts of Interest:** The authors declare no conflict of interest.

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
