# Peer review of "Thermal Modelling Utilizing Multiple Experimentally Measurable Parameters"

_batteries, doi:10.3390/batteries8100147_

Round 1
Reviewer 1 Report
This manuscript attempts to model the thermal generation associated with the internal impedance resistance of LiFePO4 batteries through various temperatures, charging rates, and different states. Three thermal models have been proposed to elucidate the internal impedance. However, there are still several issues that need to be addressed before acceptance for publication:
1) Since the purpose of the study is to model the impact of various conditions on the thermal generation inside of the battery, it is important to elaborate on the physical meaning of the data and further correlate the models to the actual condition of the battery.
2) According to the first model, the author note that the internal resistance decreased with the increasing C-rates (line 120). However, higher C-rates usually cause a high IR drop due to high internal impedance.
3) Unclear experimental descriptions, in particular in the thermal data collection: in figure 1 there were only four thermocouple sites, however, the author mentioned the data collected from 8 thermocouples (line 175). Details calculations methods or software are required in the experimental descriptions.
4) Conclusion section for the three different models needs to be added since the large and complex data collection.
5) The figure arrangement was quite confusing, e.g. the figure legend was too small (Figure 2, Figure 4, Figure 6).
6) The typography of the manuscript is messy. There is some error message “Error! Reference source not found.” in the manuscript. Please check carefully the referencing method.
Author Response
This manuscript attempts to model the thermal generation associated with the internal impedance resistance of LiFePO4 batteries through various temperatures, charging rates, and different states. Three thermal models have been proposed to elucidate the internal impedance. However, there are still several issues that need to be addressed before acceptance for publication:
- Since the purpose of the study is to model the impact of various conditions on the thermal generation inside of the battery, it is important to elaborate on the physical meaning of the data and further correlate the models to the actual condition of the battery.
Response: Thank you for this comment. I have added multiple additional physical meaning paragraphs as the reviewer has suggested. Including “We note the A2 terms are a function of the SOH while the A9 term is a function of the SOC. The last term is noted to be a compound effect of the SOC and the SOH with the temperatures effect being inside an exponential. … We note that the diffusion coefficient for the Iron Phosphate anode is a function of the exponential of the temperature from the temperature range of 120⁰C to -50⁰C extrapolating outside of which we assume a constant effect” “We see the specific heat capacity and the density increase with degradation.”
- According to the first model, the author note that the internal resistance decreased with the increasing C-rates (line 120). However, higher C-rates usually cause a high IR drop due to high internal impedance.
Response: I have added a few lines regarding the influence of C rate on the internal resistance, the overpotential and the total energy loss. I have also added a sentence about the effects outside of the experimental range closer to the current limit. “We note that the internal resistance is seen to decrease with an increase in the C rate in this current range which is still not near the current limit for the cell. The overall energy loss and the overpotential however increases as C-rate increases.”
- Unclear experimental descriptions, in particular in the thermal data collection: in figure 1 there were only four thermocouple sites, however, the author mentioned the data collected from 8 thermocouples (line 175). Details calculations methods or software are required in the experimental descriptions.
Response: I have added detailed calculations to the experimental section. The added section includes a description of using a Voronoi diagram to calculate the average surface temperature based on a surface area weighted average of batteries measured temperatures at each thermocouple location along with the equation for calculation.
- Conclusion section for the three different models needs to be added since the large and complex data collection.
Response: Conclusion section added. Thank you for your input.
- The figure arrangement was quite confusing, e.g. the figure legend was too small (Figure 2, Figure 4, Figure 6).
Response: Thank you for your comment. The figures were modified to clearly show the legends. All figures are now at 600dpi and can be modified by the journal to fit template as necessary.
- The typography of the manuscript is messy. There is some error message “Error! Reference source not found.” in the manuscript. Please check carefully the referencing method.
Response: The author has corrected all the “Error!” Messages. Thank you for your comment

Reviewer 2 Report
The authors did a very good job. However, I have few questions
1. The introduction needs to be improved and made clear on which model are the authors interested in.
2. I know you have 45 results for each case. However, can you mention which C rate and SOH gave the best R2 value. Like in abstract you mentioned the R2 for all three cases. But for which C rate and SOH?
3. Also what do you want to conclude from Figure 6? You haven't mentioned any R2 values for the 2D model, but you did mention that in abstract.
4. I feel the paper is still incomplete and needs a conclusion on what the author wants to conclude from their work.
Author Response
The authors did a very good job. However, I have few questions
- The introduction needs to be improved and made clear on which model are the authors interested in.
Response: An additional section has been added to the introduction to clarify the models that are pursued in the paper. Please see “This paper focuses on an equivalent circuit model approach that incorporates physio-chemical theory into developing a nonlinear equation for the internal resistance. Once the nonlinear model for the internal resistance is built we use a simple thermal model to simulate heating effects both from the internal resistance and secondly from the reversible heat. The thermal 2D model also incorporates tab resistances as an additional heat source term.” Thank you for your comment.
- I know you have 45 results for each case. However, can you mention which C rate and SOH gave the best R2 value. Like in abstract you mentioned the R2 for all three cases. But for which C rate and SOH
Response: Thank you for your comment. All of the highest accuracy runs have been added and discussed in the paper as suggested by reviewer. Please see “The highest accuracy Pearson values are seen to occur at 288 K, 65% SOH and 54 Amp current for the non-reversible model and at 278 K, 91% SOH and 54 Amp current for the Reversible model.” “The highest accuracy single test is at 91% SOH, 278 K and 36 Amps. The overall R2 value for the 2D thermal model was 0.996.”
- Also what do you want to conclude from Figure 6? You haven't mentioned any R2 values for the 2D model, but you did mention that in abstract.
Response: Additional section added to paper as suggested. R2 is discussed in results and conclusions section further. Please see “The table shows the model is more accurate at low temperatures than at high temperatures. The model is most accurate at 83%SOH and least accurate at 65%SOH with 91%SOH being in between. . The highest accuracy single test is at 91% SOH, 278 K and 36 Amps. The overall R2 value for the 2D thermal model was 0.996.”
- I feel the paper is still incomplete and needs a conclusion on what the author wants to conclude from their work.
Response: Conclusion section added to paper. Thank you for your comment

Reviewer 3 Report
In this manuscript, the authors carried out a simulation experiment for LiFePO4 based battery cell using equivalent circuit modals with multiple input parameters such as State of Health (SOH), State of Charge (SOC), Current and Temperature. The attempted motivation of this work is encouraging, but the manuscript lacks a fundamental insight in terms of electrochemistry and fundamental battery materials with addressing practical perspective. Therefore, based on the present format, the manuscript cannot be accepted for publication in the Batteries journal.
Some comments are given below
1. The problem statement and methods are not well described in the manuscript. The tables and figures are not numbered properly in the manuscript, and it is confusing a lot for the reviewer to address which figure conveys what information.
2. In terms of experiments, the authors mentioned that the simulation experiments agree with experimental results in some part of the manuscript. The section is not referenced properly, and it must be addressed with appropriate references.
3. A huge data set, tables and figures need to be organized carefully.
4. This manuscript doesn’t express the appropriate conclusion what the authors observed.
Considering my expertise, I am willing to reject this manuscript in the current form. This article needs a strong revision before submitting to any other journal.
Author Response
- In terms of experiments, the authors mentioned that the simulation experiments agree with experimental results in some part of the manuscript. The section is not referenced properly, and it must be addressed with appropriate references.
Response: Thank you for your comment. All “Reference not found” errors have been corrected and point to the relevant data that is being referenced. In addition to this references have been added including “Mevawalla, A., Panchal, S., Tran, M., Fowler, M., & Fraser, R. (2020). Mathematical Heat Transfer Modeling and Experimental Validation of Lithium-Ion Battery…”
- A huge data set, tables and figures need to be organized carefully.
Response: Thank you for your comment. I have reorganized the entire paper as the reviewer has suggested to clearly present the data and results. The author notes that the manuscript had no error messages and organized to standard before the reorganization into MDPI template not on my end. Since then the manuscript has been reorganized by me in the new template form.
- This manuscript doesn’t express the appropriate conclusion what the authors observed.
Response: A conclusion section has been added as the author has suggested. Thank you for your input.

Round 2
Reviewer 1 Report
The authors have revised the manuscript according to the comments. However, minor corrections should be made, for example on line 146. Why there is an Iron phosphate anode? according to Table 1, the anode of the battery was graphite.
Author Response
Thank you for your comment. The author has revised this error to state "graphite anode".

Reviewer 2 Report
Thank you for editing the manuscript. I still feel the manuscript needs a bit more work to look presentable, before it can be published.
1. The author mentioned "Voronoi diagram". What is Voronoi diagram? How is the author using it? Can you please briefly explain in a couple of lines what is the significance of this, what are those points on the figure and how is the author using it?
2. I still feel confused looking at so many graphs and to figure out which one is the best or even the tables. Can you highlight the best value in the table and also the best graph to make it easy?
3. I feel the paper still needs a proper detailed conclusion, not just the R2 squared values.
4. Also for the decimal values in tables, can you round them off to 2 or 3 decimal points so it won't look like a lot of data. Since you already have a lot of data such big decimal numbers is making it look scary.
Author Response
1. The author mentioned "Voronoi diagram". What is Voronoi diagram? How is the author using it? Can you please briefly explain in a couple of lines what is the significance of this, what are those points on the figure and how is the author using it?
Thank you for your comment. The author has added the following regarding the issue "The points shown in Figure 2 show the locations of the thermocouples on the battery surface. The use of the Voronoi diagram ensures that the area assigned to each thermocouple is the area that is truly closest to the sensor’s point on the surface and is not simply an arbitrary area assigned to that weighted measurement. By doing this the appropriate magnitude of surface area is assigned to each thermocouple’s experimental measurement. This confirms the single point average surface area temperature is a legitimate measurement of the true battery surface temperature."
2. I still feel confused looking at so many graphs and to figure out which one is the best or even the tables. Can you highlight the best value in the table and also the best graph to make it easy?
Thank you for your comment. The author has done as the comment suggests and highlighted the key points in both the table and figures.
3. I feel the paper still needs a proper detailed conclusion, not just the R2 squared values.
As the reviewer has suggested the conclusion has been modified and extended. Please see "The purpose of the paper was to create models that utilize practically measurable parameters to simulate a lithium ion battery’s internal resistance and surface temperature. Overall we see that all models have a high accuracy with the data. Equation 3 shows the internal resistance as the sum of 5 terms where the first term is a simple constant. We see a term with asinh of both the current and SOC expected to be related to the activation overpotential and a term with an exponential of the reciprocal of temperature expected to be related to mass transfer losses. The term is is multiplied by the A9 parameter which is a function of the SOC being zero at SOC’s above 35% contributing little to the internal resistance.
The internal resistance model is then used in 3 thermal models; two 0D models and one 2D Finite Element model. The 0D models are with and without reversible heat terms. The reversible heat is calculated using the Bernardi heat equation. The 2D model also adds tab heating junction resistance as heat source terms for each tab."
4. Also for the decimal values in tables, can you round them off to 2 or 3 decimal points so it won't look like a lot of data. Since you already have a lot of data such big decimal numbers is making it look scary.
Thank you for your comment. As requested the tables have been rounded to 3 decimal points and reorganized to a more clear format.

Reviewer 3 Report
Thanks for the revised manuscript.Even though the authors clarified some of the concerns in previous review, the manuscript still needs a revision especially the conclusion and methods section.
The manuscript has a huge data set including graphs and tables, but the conclusion part is not appropriately conveying the overall objective and outcome of the results.The authors please consider revising the manuscript again before submitting to other journal.
Thanks.
Author Response
Thanks for the revised manuscript. Even though the authors clarified some of the concerns in previous review, the manuscript still needs a revision especially the conclusion and methods section.
The manuscript has a huge data set including graphs and tables, but the conclusion part is not appropriately conveying the overall objective and outcome of the results. The authors please consider revising the manuscript again before submitting to other journal.
Thanks.
Thank you for your comment. As the Reviewer has suggested the author has modified both the conclusion and the experimental methods section. Please see the following:
Methods: "This paper focuses on an equivalent circuit model approach that incorporates physio-chemical theory into developing a nonlinear equation for the internal resistance. Once the nonlinear model for the internal resistance is built we use a simple thermal model to simulate heating effects both from the internal resistance and secondly from the reversible heat. The thermal 2D model also incorporates tab resistances as an additional heat source term. ....
The back of the battery has the thermocouples in the same positions as the front of the battery with thermocouple 5 being on the anode, 6 being on the cathode, 7 in the center and 8 on the bottom
The points shown in Figure 2 show the locations of the thermocouples on the battery surface. The use of the Voronoi diagram ensures that the area assigned to each thermocouple is the area that is truly closest to the sensor’s point on the surface and is not simply an arbitrary area assigned to that weighted measurement. By doing this the appropriate magnitude of surface area is assigned to each thermocouple’s experimental measurement. This confirms the single point average surface area temperature is a legitimate measurement of the true battery surface temperature."
Conclusion: "The purpose of the paper was to create models that utilize practically measurable parameters to simulate a lithium ion battery’s internal resistance and surface temperature. Overall we see that all models have a high accuracy with the data. Equation 3 shows the internal resistance as the sum of 5 terms where the first term is a simple constant. We see a term with asinh of both the current and SOC expected to be related to the activation overpotential and a term with an exponential of the reciprocal of temperature expected to be related to mass transfer losses. The term is is multiplied by the A9 parameter which is a function of the SOC being zero at SOC’s above 35% contributing little to the internal resistance.
The internal resistance model is then used in 3 thermal models; two 0D models and one 2D Finite Element model. The 0D models are with and without reversible heat terms. The reversible heat is calculated using the Bernardi heat equation. The 2D model also adds tab heating junction resistance as heat source terms for each tab."

Round 3
Reviewer 1 Report
No further revision required
Reviewer 2 Report
1. The best result is not highlighted in Figure 5 (b)
2. In page 21 the graph on top right is missing the current and SOH values
3. You need to label Figure 7 properly. You just labeled (a), (b) and (c). It should be Figure 7 (a), 7 (b) and 7(c).
4. Also is there a reason why you are not highlighting the best results in Figure 7 (a) and 7 (b)?
5. Please check the tables again not all decimal values are rounded off. Please revise the whole paper for typos and mistakes.
Author Response
1. The best result is not highlighted in Figure 5 (b)
I have highlighted the figure as reviewer has suggested.
2. In page 21 the graph on top right is missing the current and SOH values
Thank you for your comment. The error has been corrected and the graph now contains the appropriate information.
3. You need to label Figure 7 properly. You just labeled (a), (b) and (c). It should be Figure 7 (a), 7 (b) and 7(c).
Thank you for your comment. The error has been corrected.
4. Also is there a reason why you are not highlighting the best results in Figure 7 (a) and 7 (b)?
Thank you for your comment. The author has highlighted as reviewer suggested.
5. Please check the tables again not all decimal values are rounded off. Please revise the whole paper for typos and mistakes.
Thank you for your comment. The author has carefully gone through the paper and rounded off all R values along with other values and looked thoroughly for typos and errors correcting them immediately.

Reviewer 3 Report
Thanks to authors for their careful revision.
This article still needs improvement in terms of highlighting appropriate results in a comprehensive way.This article has huge data sets including tables and figures, it is difficult to follow in many cases.
Please try to highlight the appropriate results with an associated text.
Thanks.
Author Response
Thank you for your comment. The author has thoroughly revised the paper and highlighted appropriate results and corrected errors as the reviewer has suggested. The author has reorganized the paper to more clearly present the large amount of data being shown so it is easier to follow.

Round 4
Reviewer 2 Report
Thanks for addressing the comments.
Reviewer 3 Report
Thanks to authors for the review. However, the authors could have presented the results and review changes much better way.
Thanks.